# Diffusion Models Beat GANs on Image Synthesis

**Prafulla Dhariwal**[*]
OpenAI
prafulla@openai.com

**Alex Nichol**[*]
OpenAI
alex@openai.com

## Abstract

We show that diffusion models can achieve image sample quality superior to the current state-of-the-art generative models. We achieve this on unconditional image synthesis by finding a better architecture through a series of ablations. For conditional image synthesis, we further improve sample quality with classifier guidance: a simple, compute-efficient method for trading off diversity for fidelity using gradients from a classifier. We achieve an FID of 2.97 on ImageNet 128×128, 4.59 on ImageNet 256×256, and 7.72 on ImageNet 512×512, and we match BigGAN-deep even with as few as 25 forward passes per sample, all while maintaining better coverage of the distribution. Finally, we find that classifier guidance combines well with upsampling diffusion models, further improving FID to 3.94 on ImageNet 256×256 and 3.85 on ImageNet 512×512.

## 1   Introduction

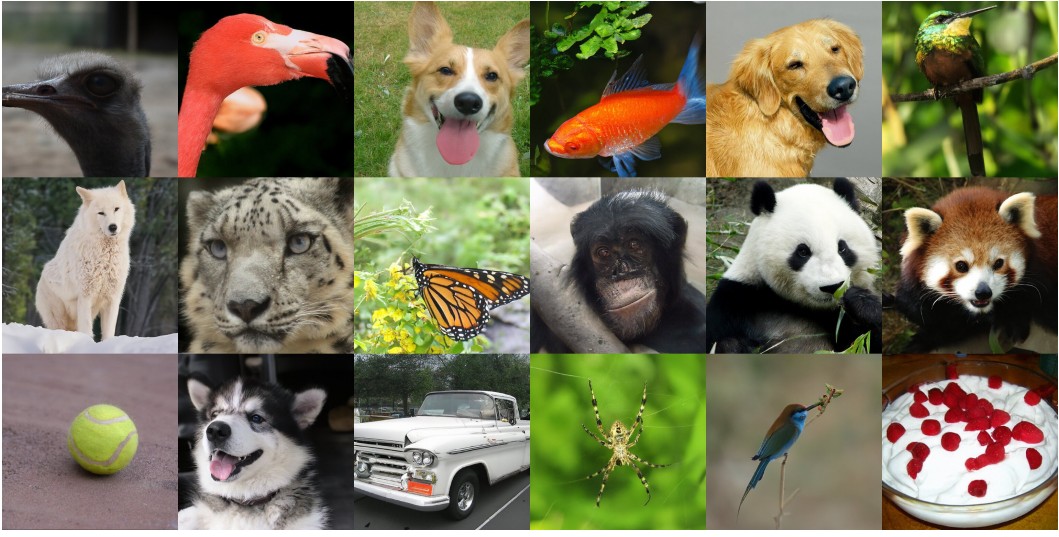

Figure 1: Selected samples from our best ImageNet 512×512 model (FID 3.85)

Over the past few years, generative models have gained the ability to generate human-like natural language [9], high-quality synthetic images [8, 34, 57] and highly diverse human speech and music [70, 17]. These models can be used in a variety of ways, such as generating images from text prompts [78, 56] or learning useful feature representations [18, 10]. While these models are already capable

---

[*]Equal contribution

35th Conference on Neural Information Processing Systems (NeurIPS 2021).

of producing realistic images and sound, there is still much room for improvement beyond the current state-of-the-art, and better generative models could have wide-ranging impacts on graphic design, games, music production, and countless other fields.

GANs [25] currently hold the state-of-the-art on most image generation tasks [8, 74, 34] as measured by sample quality metrics such as FID [29], Inception Score [61] and Precision [38]. However, some of these metrics do not fully capture diversity, and it has been shown that GANs capture less diversity than state-of-the-art likelihood-based models [57, 49, 48]. Furthermore, GANs are often difficult to train, collapsing without carefully selected hyperparameters and regularizers [8, 47, 7]. While GANs hold the state-of-the-art, their drawbacks make them difficult to scale and apply to new domains. As a result, much work has been done to achieve GAN-like sample quality with likelihood-based models [22, 57, 31, 48, 12]. While these models capture more diversity and are typically easier to scale and train than GANs, they still fall short in terms of visual fidelity. Furthermore, except for VAEs, sampling from these models is slower than GANs in terms of wall-clock time.

Diffusion models are a class of likelihood-based models which have recently been shown to produce high-quality images [63, 66, 31, 49] while offering desirable properties such as distribution coverage, a stationary training objective, and easy scalability. These models generate samples by gradually removing noise from a signal, and their training objective can be expressed as a reweighted variational lower-bound [31]. This class of models already holds the state-of-the-art [67] on CIFAR-10 [37], but still lags behind GANs on difficult generation datasets like LSUN and ImageNet. We hypothesize that this gap exists for at least two reasons: first, that the model architectures used by recent GAN literature have been heavily explored and refined; second, that GANs are able to trade off diversity for fidelity, producing high quality samples but not covering the whole distribution. We aim to bring these benefits to diffusion models, first by improving model architecture and then by devising a scheme for trading off diversity for fidelity.

The rest of the paper is organized as follows. In Section 2, we give a brief background of diffusion models based on Ho et al. [31] and the improvements from Nichol and Dhariwal [49] and Song et al. [64], and we describe our evaluation setup. In Section 3, we introduce simple architecture improvements that give a substantial boost to FID. In Section 4, we describe a method for using gradients from a classifier to guide a diffusion model during sampling. Finally, in Section 5 we show that models with our improved architecture achieve state-of-the-art on unconditional image synthesis tasks, and with classifier guidance achieve state-of-the-art on conditional image synthesis.

## 2 Background

In this section, we provide a brief overview of diffusion models. For a more detailed mathematical description, we refer the reader to Appendix C. On a high level, diffusion models sample from a distribution by reversing a gradual noising process. In particular, sampling starts with noise $x_T$ and produces gradually less-noisy samples $x_{T-1}, x_{T-2}, ...$ until reaching a final sample $x_0$. In particular, a diffusion model learns to produce a slightly more "denoised" $x_{t-1}$ from $x_t$. Ho et al. [31] parameterize this model using a function $\epsilon_\theta(x_t, t)$ which predicts the noise component of a noisy sample $x_t$. To train this function, each sample in a minibatch is produced by randomly drawing a data sample $x_0$, a timestep $t$, and noise $\epsilon$, which together give rise to a noised sample $x_t$ (Equation 3, Appendix C). The training objective is then $||\epsilon_\theta(x_t, t) - \epsilon||^2$, i.e. a simple mean-squared error loss between the true noise and the predicted noise (Equation 12, Appendix C).

Ho et al. [31] show that, under reasonable assumptions, we can then model the denoising distribution $p_\theta(x_{t-1}|x_t)$ of $x_{t-1}$ given $x_t$ as a diagonal Gaussian $\mathcal{N}(x_{t-1}; \mu_\theta(x_t, t), \Sigma_\theta(x_t, t))$, where the mean $\mu_\theta(x_t, t)$ can be calculated as a function of $\epsilon_\theta(x_t, t)$ (Equation 13, Appendix C). Ho et al. [31] observe that the simple mean-squared error objective, $L_{\text{simple}}$, works better in practice than the actual variational lower bound $L_{\text{vlb}}$ that can be derived from interpreting the denoising diffusion model as a VAE. They also note that training with this objective and using their corresponding sampling procedure is equivalent to the denoising score matching model from Song and Ermon [65], who use Langevin dynamics to sample from a denoising model trained with multiple noise levels to produce high quality image samples. We often use "diffusion models" as shorthand to refer to both classes of models.

Following the breakthrough work of Song and Ermon [65] and Ho et al. [31], several recent papers have proposed improvements to diffusion models. Nichol and Dhariwal [49] find that fixing

the variance $\Sigma_\theta(x_t, t)$ to a constant as done in Ho et al. [31] is sub-optimal for sampling with fewer diffusion steps, and propose to parameterize $\Sigma_\theta(x_t, t)$ as a neural network whose output $v$ is interpolated as $\Sigma_\theta(x_t, t) = \exp(v \log \beta_t + (1 - v) \log \tilde{\beta}_t)$. Here, $\beta_t$ and $\tilde{\beta}_t$ (Equation 5, Appendix C) are the variances in Ho et al. [31] corresponding to upper and lower bounds for the reverse process variances. Additionally, Nichol and Dhariwal [49] propose a hybrid objective for training both $\epsilon_\theta(x_t, t)$ and $\Sigma_\theta(x_t, t)$ using the weighted sum $L_{\text{simple}} + \lambda L_{\text{vlb}}$. Learning the reverse process variances with their hybrid objective allows sampling with fewer steps without much drop in sample quality. We adopt this objective and parameterization, and use it throughout our experiments.

Song et al. [64] propose DDIM, which formulates an alternative non-Markovian noising process that has the same forward marginals as DDPM, but allows producing different reverse samplers by changing the variance of the reverse noise. By setting this noise to 0, they provide a way to turn any model $\epsilon_\theta(x_t, t)$ into a deterministic mapping from latents to images, and find that this provides an alternative way to sample with fewer steps. We adopt this sampling approach when using fewer than 50 sampling steps, since Nichol and Dhariwal [49] found it to be beneficial in this regime.

**Sample Quality Metrics**: For comparing sample quality across models, we perform quantitative evaluations using the following metrics. While these metrics are often used in practice and correspond well with human judgement, they are not a perfect proxy, and finding better metrics for sample quality evaluation is still an open problem.

We use FID [29] as our default metric for overall sample quality comparisons as it captures both fidelity and diversity and has been the de facto standard metric for state-of-the-art generative models [33, 34, 8, 31]. We use Precision and Recall [38] as proxies for separately measuring fidelity and diversity, respectively. We include sFID [48] as a metric that better captures spatial relationships than FID, and also include Inception Score (IS) [61] as another proxy for fidelity. When comparing against other methods, we re-compute these metrics using public samples or models whenever possible. This is for two reasons: first, some papers [33, 34, 31] compare against arbitrary subsets of the training set which are not readily available; and second, subtle implementation differences can affect the resulting FID values [51]. For consistent comparisons, we use the full training set as the reference batch [29, 8], and evaluate metrics for all models using the same codebase.

## 3 Architecture Improvements

Ho et al. [31] adopted the UNet architecture [58] for diffusion models, which Jolicoeur-Martineau et al. [32] found to substantially improve sample quality over the previous architectures [65, 39] used for denoising score matching. The UNet model uses a stack of residual layers and downsampling convolutions, followed by a stack of residual layers with upsampling convolutions, with skip connections connecting the layers with the same spatial size. In addition, they use a global attention layer at the 16×16 resolution with a single head, and add a projection of the timestep embedding into each residual block. Song et al. [67] found that further changes to the UNet architecture improved performance on the CIFAR-10 [37] and CelebA-64 [40] datasets. We show the same result on ImageNet 128×128, finding that architecture can indeed give a substantial boost to sample quality on a much larger and more diverse datasets at a higher resolution.

We explore the following architectural changes: increasing depth versus width, holding model size relatively constant; increasing the number of attention heads; using attention at 32×32, 16×16, and 8×8 resolutions rather than only at 16×16; using the BigGAN [8] residual block for upsampling and downsampling the activations, following [67]; and finally; rescaling residual connections with $\frac{1}{\sqrt{2}}$, following [67, 33, 34].

We train models with the above architecture changes on ImageNet 128×128 and compare them on FID, evaluated at two different points of training, in Table 1. Aside from rescaling residual connections, all of the other modifications improve performance and have a positive compounding effect. On wall-clock (Figure 5, Appendix A.1) we find that increased depth hurts training time most, so we opt not to use this change in further experiments. We also study other attention configurations that better match the Transformer architecture [72]. We try two configurations: constant attention heads, or constant channels per head. Table 2 shows our results, indicating that more heads or fewer channels per head improves FID. On wall-clock (Figure 5, Appendix A.1), we see that 64 channels is best so we opt to use 64 channels per head as our default. We note that this choice also better matches modern transformer architectures, and is on par with our other configurations in terms of final FID.

Table 1: Ablation of various architecture changes, evaluated at 700K and 1200K iterations

| Channels | Depth | Heads | Attention resolutions | BigGAN up/downsample | Rescale resblock | FID 700K | FID 1200K |
|---|---|---|---|---|---|---|---|
| 160 | 2 | 1 | 16 | ✗ | ✗ | 15.33 | 13.21 |
| 128 | 4 | | | | | -0.21 | -0.48 |
| | | 4 | | | | -0.54 | -0.82 |
| | | | 32,16,8 | | | -0.72 | -0.66 |
| | | | | ✓ | | -1.20 | -1.21 |
| | | | | | ✓ | 0.16 | 0.25 |
| 160 | 2 | 4 | 32,16,8 | ✓ | ✗ | **-3.14** | **-3.00** |

Table 2: Ablation of attention heads. More heads or lower channels per heads both improve FID. The base model was a smaller version of the best model from Table 1.

| Number of heads | Channels per head | FID |
|---|---|---|
| 1 | | 14.08 |
| 2 | | -0.50 |
| 4 | | -0.97 |
| 8 | | -1.17 |
| | 32 | -1.36 |
| | 64 | -1.03 |
| | 128 | -1.08 |

We also experiment with a layer [49] that we refer to as adaptive group normalization (AdaGN), which incorporates the timestep and class embedding into each residual block after a group normalization operation [75], similar to adaptive instance norm [33] and FiLM [54]. We define this layer as $\text{AdaGN}(h, y) = y_s \, \text{GroupNorm}(h) + y_b$, where $h$ is the intermediate activations of the residual block following the first convolution, and $y = [y_s, y_b]$ is obtained from a linear projection of the timestep and class embedding. We had already seen AdaGN improve our earliest diffusion models, and so had included it by default in all our runs. We explicitly ablate this choice (Table 6, Appendix A.1), and find that FID becomes worse by 2.02 when we remove the adaptive group normalization layer.

In the rest of the paper, we use this final improved model architecture as our default: variable width with 2 residual blocks per resolution, multiple heads with 64 channels per head, attention at 32, 16 and 8 resolutions, BigGAN residual blocks for up and downsampling, and adaptive group normalization for injecting timestep and class embeddings into residual blocks.

## 4   Classifier Guidance

In addition to employing well designed architectures, GANs for conditional image synthesis [45, 8] make heavy use of class labels. This often takes the form of class-conditional normalization statistics [20, 14] as well as discriminators with heads explicitly designed to behave like classifiers $p(y|x)$ [46]. As further evidence that class information is crucial to the success of these models, Lucic et al. [42] find that it is helpful to generate synthetic labels when working in a label-limited regime. Given this observation for GANs, it makes sense to explore different ways to condition diffusion models on class labels. We already incorporate class information into adaptive group normalization layers (Section 3). Here, we explore a different approach: exploiting a classifier $p(y|x)$ to improve a diffusion generator. Sohl-Dickstein et al. [63] and Song et al. [67] show one way to achieve this, wherein a pre-trained diffusion model can be conditioned using the gradients of a classifier. In particular, we can train a classifier $p_\phi(y|x_t, t)$ on noisy images $x_t$, and then use gradients $\nabla_{x_t} \log p_\phi(y|x_t, t)$ to guide the diffusion sampling process towards an arbitrary class label $y$.

For class conditional diffusion sampling, we reproduce the derivation from Sohl-Dickstein et al. [63] in Appendix D.2. For DDIM, we perform a score-based derivation in Appendix D.3 inspired by Song et al. [67]. The resulting sampling algorithms we use for guidance are Algorithms 1 and 2 respectively. Both algorithms incorporate class information by adding the gradients of a classifier to each sampling step with an appropriate step size. In these algorithms, we choose the notation

**Algorithm 1** Classifier guided diffusion sampling, given a diffusion model $(\mu_\theta(x_t), \Sigma_\theta(x_t))$, classifier $p_\phi(y|x_t)$, and gradient scale $s$.

---

**Input:** class label $y$, gradient scale $s$
$x_T \leftarrow$ sample from $\mathcal{N}(0, \mathbf{I})$
**for all** $t$ from $T$ to 1 **do**
    $\mu, \Sigma \leftarrow \mu_\theta(x_t), \Sigma_\theta(x_t)$
    $x_{t-1} \leftarrow$ sample from $\mathcal{N}(\mu + s\Sigma \nabla_{x_t} \log p_\phi(y|x_t), \Sigma)$
**end for**
**return** $x_0$

---

**Algorithm 2** Classifier guided DDIM sampling, given a diffusion model $\epsilon_\theta(x_t)$, classifier $p_\phi(y|x_t)$, and gradient scale $s$.

---

**Input:** class label $y$, gradient scale $s$
$x_T \leftarrow$ sample from $\mathcal{N}(0, \mathbf{I})$
**for all** $t$ from $T$ to 1 **do**
    $\hat{\epsilon} \leftarrow \epsilon_\theta(x_t) - \sqrt{1 - \bar{\alpha}_t} \nabla_{x_t} \log p_\phi(y|x_t)$
    $x_{t-1} \leftarrow \sqrt{\bar{\alpha}_{t-1}} \left( \frac{x_t - \sqrt{1 - \bar{\alpha}_t}\hat{\epsilon}}{\sqrt{\bar{\alpha}_t}} \right) + \sqrt{1 - \bar{\alpha}_{t-1}}\hat{\epsilon}$
**end for**
**return** $x_0$

---

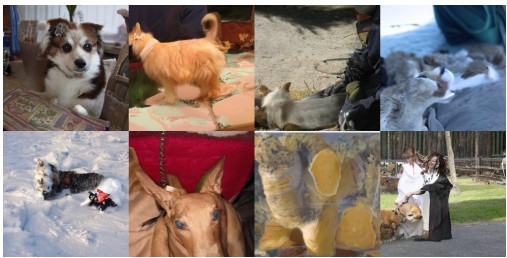 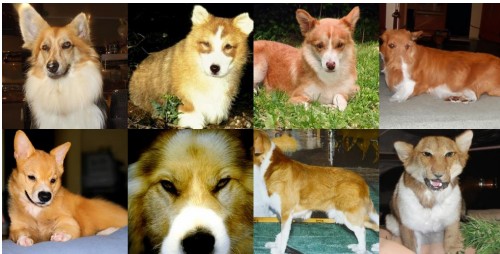

Figure 2: Samples from an unconditional diffusion model with classifier guidance to condition on the class "Pembroke Welsh corgi". Using classifier scale 1.0 (left; FID: 33.0) does not produce convincing samples in this class, whereas classifier scale 10.0 (right; FID: 12.0) produces much more class-consistent images.

$p_\phi(y|x_t, t) = p_\phi(y|x_t)$ and $\epsilon_\theta(x_t, t) = \epsilon_\theta(x_t)$ for brevity, noting that they refer to separate functions for each timestep $t$ and at training time the models must be conditioned on the input $t$.

To apply classifier guidance to a large scale generative task, we train classification models on ImageNet. Our classifier architecture is simply the downsampling trunk of the UNet model with an attention pool [55] at the 8x8 layer to produce the final output. We train these classifiers on the same noising distribution as the corresponding diffusion model, and also add random crops to reduce overfitting.

In initial experiments with unconditional ImageNet models, we found it necessary to scale the classifier gradients by a constant factor larger than 1. When using a scale of 1, we observed that the classifier assigned reasonable probabilities (around 50%) to the desired classes for the final samples, but these samples did not match the intended classes upon visual inspection. Scaling up the classifier gradients remedied this problem, and the class probabilities from the classifier increased to nearly 100%. Figure 2 shows an example of this effect. To understand the effect of scaling classifier gradients, note that $s \cdot \nabla_x \log p(y|x) = \nabla_x \log \frac{1}{Z} p(y|x)^s$, where $Z$ is an arbitrary constant. As a result, the conditioning process is still theoretically grounded in a re-normalized classifier distribution proportional to $p(y|x)^s$. When $s > 1$, this distribution becomes sharper than $p(y|x)$, since larger values are amplified by the exponent. In other words, using a larger gradient scale focuses more on the modes of the classifier, which is potentially desirable for producing higher quality (but less diverse) samples.

In the above derivations, we assumed that the underlying diffusion model was unconditional, modeling $p(x)$. It is also possible to train conditional diffusion models, $p(x|y)$, and use classifier guidance in

Table 3: Effect of classifier guidance on sample quality. Both conditional and unconditional models were trained for 2M iterations on ImageNet 256×256 with batch size 256.

| Conditional | Guidance | Scale | FID | sFID | IS | Precision | Recall |
|:---:|:---:|:---:|:---:|:---:|:---:|:---:|:---:|
| ✗ | ✗ | | 26.21 | **6.35** | 39.70 | 0.61 | 0.63 |
| ✗ | ✓ | 1.0 | 33.03 | 6.99 | 32.92 | 0.56 | **0.65** |
| ✗ | ✓ | 10.0 | **12.00** | 10.40 | **95.41** | **0.76** | 0.44 |
| ✓ | ✗ | | 10.94 | 6.02 | 100.98 | 0.69 | **0.63** |
| ✓ | ✓ | 1.0 | **4.59** | **5.25** | 186.70 | 0.82 | 0.52 |
| ✓ | ✓ | 10.0 | 9.11 | 10.93 | **283.92** | **0.88** | 0.32 |

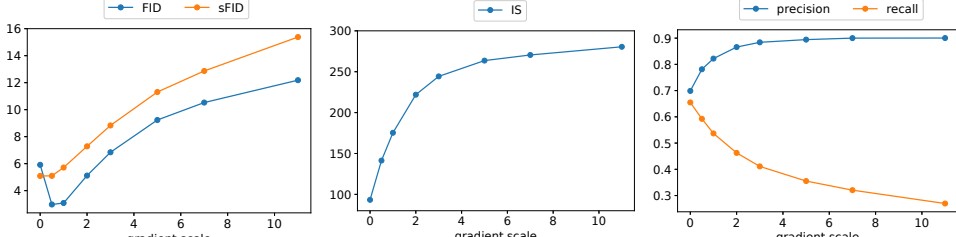

Figure 3: Change in sample quality as we vary scale of the classifier gradients for a class-conditional ImageNet 128×128 model.

the exact same way. Table 3 shows that the sample quality of both unconditional and conditional models can be greatly improved by classifier guidance. We see that, with a high enough scale, the guided unconditional model can get quite close to the FID of an unguided conditional model, although training directly with the class labels still helps. Guiding a conditional model further improves FID.

Table 3 also shows that classifier guidance improves precision at the cost of recall, thus introducing a trade-off in sample fidelity versus diversity. We explicitly evaluate how this trade-off varies with the gradient scale in Figure 3. We see that scaling the gradients beyond 1.0 smoothly trades off recall (a measure of diversity) for higher precision and IS (measures of fidelity). Since FID and sFID depend on both diversity and fidelity, their best values are obtained at an intermediate point. We also compare our guidance with the truncation trick from BigGAN (Figure 6, Appendix A.2). We find that classifier guidance is strictly better than BigGAN-deep when trading off FID for Inception Score. Less clear cut is the precision/recall trade-off, which shows that classifier guidance is only a better choice up until a certain precision threshold, after which point it cannot achieve better precision.

## 5    Results

To evaluate our improved model architecture on unconditional image generation, we train separate diffusion models on three LSUN [77] classes: bedroom, horse, and cat. To evaluate classifier guidance, we train conditional diffusion models on the ImageNet [59] dataset at 128×128, 256×256, and 512×512 resolution.

Table 4 summarizes our results. ADM refers to our **a**blated **d**iffusion **m**odel, and ADM-G additionally uses classifier **g**uidance. Our diffusion models can obtain the best FID on each task, and the best sFID on all but one task. With the improved architecture, we already obtain state-of-the-art image generation on LSUN and ImageNet 64×64. For higher resolution ImageNet, we observe that classifier guidance allows our models to substantially outperform the best GANs. These models obtain perceptual quality similar to GANs, while maintaining a higher coverage of the distribution as measured by recall, and can even do so using only 25 sampling steps. We also evaluate the computational requirements for training our models (Table 10, Appendix B), and find that we can obtain competitive sample quality while using the same or less compute than the corresponding BigGAN-deep or StyleGAN2 model.

Figure 4 compares random samples from the best BigGAN-deep model to our guided diffusion model. While the samples are of similar perceptual quality, the diffusion model contains more modes than the GAN, such as zoomed ostrich heads, single flamingos, different orientations of cheeseburgers, and a

Table 4: Sample quality comparison with state-of-the-art generative models for each task. LSUN diffusion models are sampled using 1000 steps (see Appendix L). ImageNet diffusion models are sampled using 250 steps, except when we use the DDIM sampler with 25 steps. *No BigGAN-deep model was available at this resolution, so we trained our own. †Values are taken from a previous paper, due to lack of public models or samples. ‡Results use two-resolution stacks. §Results use compute-intensive classifier rejection sampling.

| Model | FID | sFID | Prec | Rec |
|---|---|---|---|---|
| **LSUN Bedrooms 256×256** | | | | |
| DCTransformer† [48] | 6.40 | 6.66 | 0.44 | **0.56** |
| DDPM [31] | 4.89 | 9.07 | 0.60 | 0.45 |
| IDDPM [49] | 4.24 | 8.21 | 0.62 | 0.46 |
| StyleGAN [33] | 2.35 | 6.62 | 0.59 | 0.48 |
| **ADM (dropout)** | **1.90** | **5.59** | **0.66** | 0.51 |
| **LSUN Horses 256×256** | | | | |
| StyleGAN2 [34] | 3.84 | 6.46 | 0.63 | 0.48 |
| **ADM** | 2.95 | **5.94** | 0.69 | **0.55** |
| **ADM (dropout)** | **2.57** | 6.81 | **0.71** | **0.55** |
| **LSUN Cats 256×256** | | | | |
| DDPM [31] | 17.1 | 12.4 | 0.53 | 0.48 |
| StyleGAN2 [34] | 7.25 | **6.33** | 0.58 | 0.43 |
| **ADM (dropout)** | **5.57** | 6.69 | **0.63** | **0.52** |
| **ImageNet 64×64** | | | | |
| BigGAN-deep* [8] | 4.06 | 3.96 | **0.79** | 0.48 |
| IDDPM [49] | 2.92 | **3.79** | 0.74 | 0.62 |
| **ADM** | 2.61 | **3.77** | 0.73 | 0.63 |
| **ADM (dropout)** | **2.07** | 4.29 | 0.74 | **0.63** |

| Model | FID | sFID | Prec | Rec |
|---|---|---|---|---|
| **ImageNet 128×128** | | | | |
| BigGAN-deep [8] | 6.02 | 7.18 | **0.86** | 0.35 |
| LOGAN† [74] | 3.36 | | | |
| **ADM** | 5.91 | **5.09** | 0.70 | **0.65** |
| **ADM-G (25 steps)** | 5.98 | 7.04 | 0.78 | 0.51 |
| **ADM-G** | **2.97** | 5.09 | 0.78 | 0.59 |
| **ImageNet 256×256** | | | | |
| DCTransformer† [48] | 36.51 | 8.24 | 0.36 | **0.67** |
| VQ-VAE-2†‡ [57] | 31.11 | 17.38 | 0.36 | 0.57 |
| VQ-VAE-2 (RS)†‡§ [57] | ∼ 10 | | | |
| VQ-GAN‡ [21] | 15.97 | 19.05 | 0.63 | 0.58 |
| VQ-GAN (RS)‡§ [21] | 5.06 | 7.34 | 0.79 | 0.48 |
| IDDPM‡ [49] | 12.26 | 5.42 | 0.70 | 0.62 |
| SR3†‡ [60] | 11.30 | | | |
| BigGAN-deep [8] | 6.95 | 7.36 | **0.87** | 0.28 |
| **ADM** | 10.94 | 6.02 | 0.69 | 0.63 |
| **ADM-G (25 steps)** | 5.44 | 5.32 | 0.81 | 0.49 |
| **ADM-G** | **4.59** | **5.25** | 0.82 | 0.52 |
| **ImageNet 512×512** | | | | |
| BigGAN-deep [8] | 8.43 | 8.13 | **0.88** | 0.29 |
| **ADM** | 23.24 | 10.19 | 0.73 | **0.60** |
| **ADM-G (25 steps)** | 8.41 | 9.67 | 0.83 | 0.47 |
| **ADM-G** | **7.72** | **6.57** | 0.87 | 0.42 |

tinca fish with no human holding it. We also check our generated samples for nearest neighbors in the Inception-V3 feature space in Appendix E, and we show additional samples in Appendices M–O.

We also compare guidance to using a two-stage upsampling stack. Nichol and Dhariwal [49] and Saharia et al. [60] train two-stage diffusion models by combining a low-resolution diffusion model with a corresponding upsampling diffusion model. In this approach, the upsampling model is trained to upsample images from the training set, and conditions on low-resolution images that are concatenated channel-wise to the model input using a simple interpolation (e.g. bilinear). During sampling, the low-resolution model produces a sample, and then the upsampling model is conditioned on this sample. This greatly improves FID on ImageNet 256×256, but does not reach the same performance as state-of-the-art models like BigGAN-deep [49, 60], as seen in Table 4.

In Table 5, we show that guidance and upsampling improve sample quality along different axes. We use the **u**psampling stack from Nichol and Dhariwal [49] combined with our architecture improvements, which we refer to as ADM-U. While upsampling improves precision while keeping a high recall, guidance provides a knob to trade off diversity for much higher precision. We achieve the best FIDs by using guidance at a lower resolution before upsampling to a higher resolution, indicating that these approaches complement one another.

# 6 Related Work

Score based generative models were introduced by Song and Ermon [66] as a way of modeling a data distribution using its gradients, and then sampling using Langevin dynamics [73]. Ho et al. [31] found a connection between this method and diffusion models [63], and achieved excellent sample quality by leveraging this connection. After this breakthrough work, many works followed up with

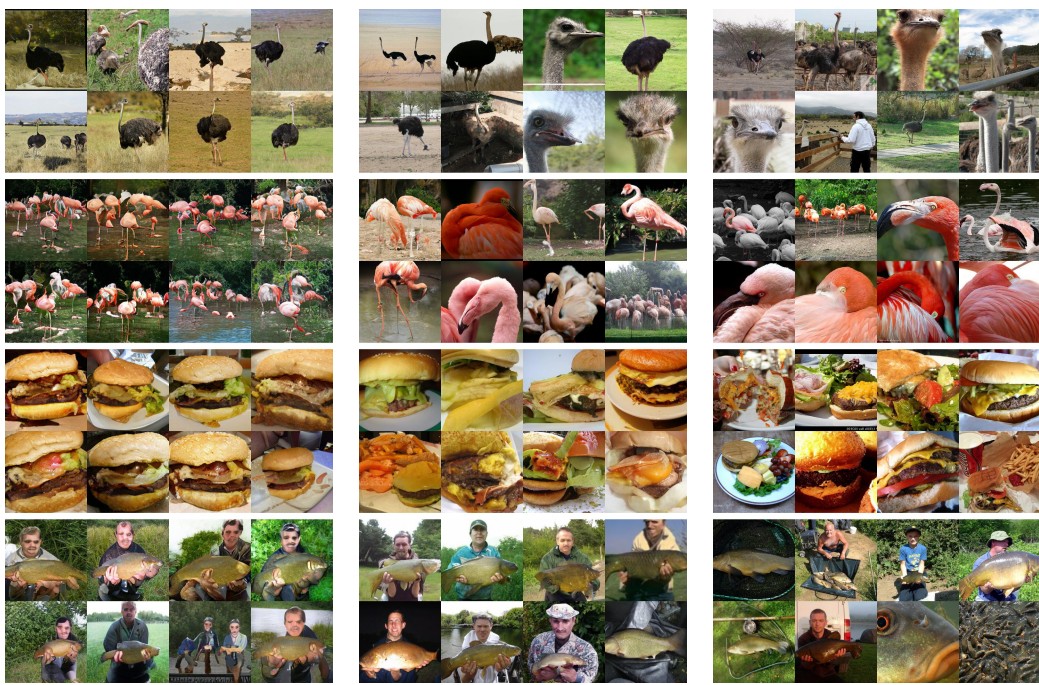

Figure 4: Samples from BigGAN-deep with truncation 1.0 (FID 6.95, left) vs samples from our diffusion model with guidance (FID 4.59, middle) and samples from the training set (right).

Table 5: Comparing our single, upsampling and classifier guided models. The upsamplers are 64→256 and 128→512. When combining guidance with upsampling, we only guide the lower resolution model. All models are sampled using 250 sampling steps.

| Model | FID | sFID | IS | Prec | Rec | Model | FID | sFID | IS | Prec | Rec |
|---|---|---|---|---|---|---|---|---|---|---|---|
| **ImageNet 256×256** | | | | | | **ImageNet 512×512** | | | | | |
| ADM | 10.94 | 6.02 | 100.98 | 0.69 | **0.63** | ADM | 23.24 | 10.19 | 58.06 | 0.73 | 0.60 |
| ADM, ADM-U | 7.49 | **5.13** | 127.49 | 0.72 | **0.63** | ADM, ADM-U | 9.96 | **5.62** | 121.78 | 0.75 | **0.64** |
| ADM-G | 4.59 | 5.25 | 186.70 | 0.82 | 0.52 | ADM-G | 7.72 | 6.57 | 172.71 | **0.87** | 0.42 |
| ADM-G, ADM-U | **3.94** | 6.14 | **215.84** | **0.83** | 0.53 | ADM-G, ADM-U | **3.85** | 5.86 | **221.72** | 0.84 | 0.53 |

more promising results: Kong et al. [36] and Chen et al. [11] demonstrated that diffusion models work well for audio; Jolicoeur-Martineau et al. [32] found that a GAN-like setup could improve samples from these models; Song et al. [67] explored ways to leverage techniques from stochastic differential equations to improve the sample quality obtained by score-based models; Song et al. [64] and Nichol and Dhariwal [49] proposed methods to improve sampling speed; Nichol and Dhariwal [49] and Saharia et al. [60] demonstrated promising results on the difficult ImageNet generation task using upsampling diffusion models. Also related to diffusion models, and following the work of Sohl-Dickstein et al. [63], Goyal et al. [27] described a technique for learning a model with learned iterative generation steps, and found that it could achieve good image samples when trained with a likelihood objective.

One missing element from previous work on diffusion models is a way to trade off diversity for fidelity. Other generative techniques provide natural levers for this trade-off. Brock et al. [8] introduced the truncation trick for GANs, wherein the latent vector is sampled from a truncated normal distribution. They found that increasing truncation naturally led to a decrease in diversity but an increase in fidelity. More recently, Razavi et al. [57] proposed to use classifier rejection sampling to filter out bad samples from an autoregressive likelihood-based model, and found that this technique improved FID. DeVries et al. [16] found that filtering out low-density regions of the training set improves GAN training performance. Most likelihood-based models also allow for low-temperature sampling [1], which provides a natural way to emphasize modes of the data distribution (see Appendix I).

Other likelihood-based models have been shown to produce high-fidelity image samples. VQ-VAE [71] and VQ-VAE-2 [57] are autoregressive models trained on top of quantized latent codes, greatly reducing the computational resources required to train these models on large images. These models produce diverse and high quality images, but still fall short of GANs without expensive rejection sampling and special metrics to compensate for blurriness. DCTransformer [48] is a related method which relies on a more intelligent compression scheme. VAEs are another promising class of likelihood-based models, and recent methods such as NVAE [69] and VDVAE [12] have successfully been applied to difficult image generation domains. Energy-based models are another class of likelihood-based models with a rich history [1, 13, 30]. Sampling from the EBM distribution is challenging, and Xie et al. [76] demonstrate that Langevin dynamics can be used to sample coherent images from these models. Du and Mordatch [19] further improve upon this approach, obtaining high quality images. More recently, Gao et al. [24] incorporate diffusion steps into an energy-based model, and find that doing so improves image samples from these models.

Other works have controlled generative models with a pre-trained classifier. For example, an emerging body of work [23, 53, 2] aims to optimize GAN latent spaces for text prompts using pre-trained CLIP [55] models. More similar to our work, Song et al. [67] uses a classifier to generate class-conditional CIFAR-10 images with a diffusion model. In some cases, classifiers can act as stand-alone generative models. For example, Santurkar et al. [62] demonstrate that a robust image classifier can be used as a stand-alone generative model, and Grathwohl et al. [28] train a model which is jointly a classifier and an energy-based model.

# 7   Limitations and Future Work

While we believe diffusion models are an extremely promising direction for generative modeling, they are still slower than GANs at sampling time due to the use of multiple denoising steps (and therefore forward passes). Since our diffusion models are also larger than the competing GAN generators, each forward pass takes anywhere from 5-20 times longer too. A promising direction to reduce this latency gap is Luhman and Luhman [43], who explore a way to distill the DDIM sampling process into a single step model. The samples from the single step model are not yet competitive with GANs, but are much better than previous single-step likelihood-based models. Future work in this direction might be able to completely close the sampling speed gap between diffusion models and GANs without sacrificing image quality.

Unlike GANs, Flows, and VAEs, diffusion models do not learn an explicit latent representation. While DDIM provides a way to encode images into an implicit latent space, it is unclear how semantically meaningful this latent representation is compared to those of other model classes. This could make it difficult to use diffusion models for representation learning or image editing applications.

The effectiveness of classifier guidance demonstrates that we can obtain powerful generative models from the gradients of a classification function. This could be used to condition an image generator with a text caption using a noisy version of CLIP [55], similar to recent methods that guide GANs using text prompts [23, 53, 2]. Our proposed classifier guidance technique is currently limited to labeled datasets. In the future, our method could be extended to unlabeled data by clustering samples to produce synthetic labels [42] or by training discriminative models to use for guidance. This also suggests that large unlabeled datasets could be leveraged in the future to pre-train powerful diffusion models that can later be improved by using a classifier with desirable properties.

# 8   Societal Impact

Our proposed technique makes generative models more accessible in terms of compute costs, especially because new classifiers can be trained and used on top of existing high-quality diffusion models. While we believe this is generally a benefit of these models, it could also have negative societal implications. For example, cheaper generative models could enable bad actors to generate fake news, propaganda images, or doctored photos. Additionally, the wide-spread deployment of these models could displace jobs in art, graphic design, animation, and photography. One could imagine, however, that democratizing generative models could also have positive impacts in the long run, creating new types of jobs such as generative photo editing. Intentionally deceitful generated

images are a more direct concern, and detecting and mitigating propaganda and fake news based on generative models is an ongoing area of research [4, 3, 5].

## 9 Conclusion

We have shown that diffusion models, a class of likelihood-based models with a stationary training objective, can obtain better sample quality than state-of-the-art GANs. Our improved architecture is sufficient to achieve this on unconditional image generation tasks, and our classifier guidance technique allows us to do so on class-conditional tasks. In the latter case, we find that the scale of the classifier gradients can be adjusted to trade off diversity for fidelity. These guided diffusion models can reduce the sampling time gap between GANs and diffusion models, although diffusion models still require multiple forward passes during sampling. Finally, by combining guidance with upsampling, we can further improve sample quality on high-resolution conditional image synthesis.

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
