# A Additional Results

## A.1 Architecture Ablations

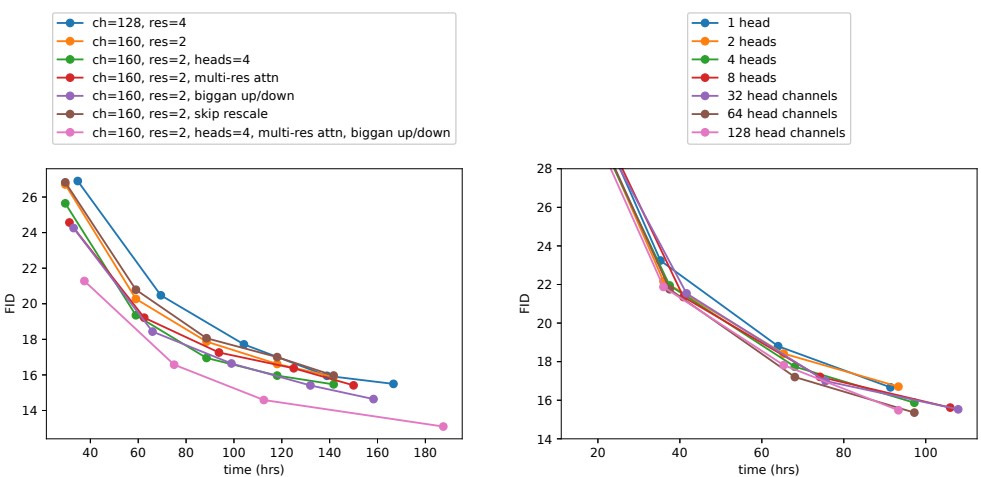

Figure 5: Ablation of various architecture changes, showing FID as a function of wall-clock time. FID evaluated over 10k samples instead of 50k for efficiency.

Table 6: Ablating the element-wise operation used when projecting timestep and class embeddings into each residual block. Replacing AdaGN with the Addition + GroupNorm layer from Ho et al. [31] makes FID worse.

| Operation | FID |
|---|---|
| Addition + GroupNorm | 15.08 |
| AdaGN | **13.06** |

## A.2 Guidance

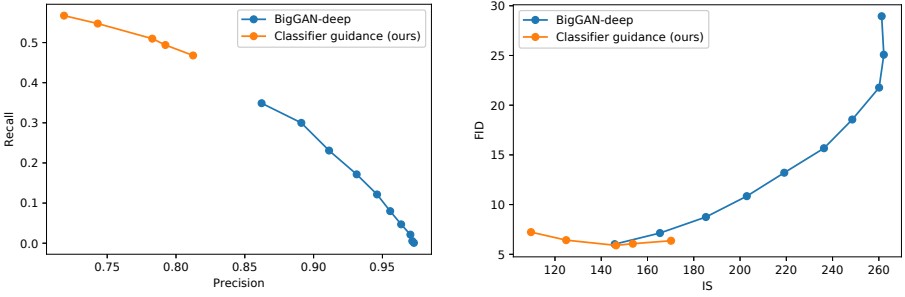

Figure 6: Trade-offs when varying truncation for BigGAN-deep and gradient scale for classifier guidance. Models are evaluated on ImageNet $128\times128$. The BigGAN-deep results were produced using the TFHub model [15] at truncation levels $[0.1, 0.2, 0.3, ..., 1.0]$.

# B    Computational Requirements

Compute is essential to modern machine learning applications, and more compute typically yields better results. It is thus important to compare our method's compute requirements to competing methods. In this section, we demonstrate that we can achieve results better than StyleGAN2 and BigGAN-deep with the same or lower compute budget.

## B.1    Throughput

We first benchmark the throughput of our models in Table 7. For the theoretical throughput, we measure the theoretical FLOPs for our model using THOP [79], and assume 100% utilization of an NVIDIA Tesla V100 (120 TFLOPs), while for the actual throughput we use measured wall-clock time. We include communication time across two machines whenever our training batch size doesn't fit on a single machine, where each of our machines has 8 V100s.

We find that a naive implementation of our models in PyTorch 1.7 is very inefficient, utilizing only 20-30% of the hardware. We also benchmark our optimized version, which use larger per-GPU batch sizes, fused GroupNorm-Swish and fused Adam CUDA ops. For our ImageNet 128×128 model in particular, we find that we can increase the per-GPU batch size from 4 to 32 while still fitting in GPU memory, and this makes a large utilization difference. Our implementation is still far from optimal, and further optimizations should allow us to reach higher levels of utilization.

Table 7: Throughput of our ImageNet models, measured in Images per V100-sec.

| Model | Implementation | Batch Size per GPU | Throughput Imgs per V100-sec | Utilization |
|---|---|---|---|---|
| 64×64 | Theoretical | - | 182.3 | 100% |
| | Naive | 32 | 37.0 | 20% |
| | Optimized | 96 | 74.1 | 41% |
| 128×128 | Theoretical | - | 65.2 | 100% |
| | Naive | 4 | 11.5 | 18% |
| | Optimized | 32 | 24.8 | 38% |
| 256×256 | Theoretical | - | 17.9 | 100% |
| | Naive | 4 | 4.4 | 25% |
| | Optimized | 8 | 6.4 | 36% |
| 64 → 256 | Theoretical | - | 31.7 | 100% |
| | Naive | 4 | 6.3 | 20% |
| | Optimized | 12 | 9.5 | 30% |
| 128 → 512 | Theoretical | - | 8.0 | 100% |
| | Naive | 2 | 1.9 | 24% |
| | Optimized | 2 | 2.3 | 29% |

## B.2    Early Stopping

In addition, we can train for many fewer iterations while maintaining sample quality superior to BigGAN-deep. Table 8 and 9 evaluate our ImageNet 128×128 and 256×256 models throughout training. We can see that the ImageNet 128×128 model beats BigGAN-deep's FID (6.02) after 500K training iterations, only one eighth of the way through training. Similarly, the ImageNet 256×256 model beats BigGAN-deep after 750K iterations, roughly a third of the way through training.

Table 8: Evaluating an ImageNet 128×128 model throughout training (classifier scale 1.0).

| Iterations | FID | sFID | Precision | Recall |
|---|---|---|---|---|
| 250K | 7.97 | 6.48 | 0.80 | 0.50 |
| 500K | 5.31 | 5.97 | 0.83 | 0.49 |
| 1000K | 4.10 | 5.80 | 0.81 | 0.51 |
| 2000K | 3.42 | 5.69 | 0.83 | 0.53 |
| 4360K | 3.09 | 5.59 | 0.82 | 0.54 |

Table 9: Evaluating an ImageNet 256×256 model throughout training (classifier scale 1.0).

| Iterations | FID | sFID | Precision | Recall |
|---|---|---|---|---|
| 250K | 12.21 | 6.15 | 0.78 | 0.50 |
| 500K | 7.95 | 5.51 | 0.81 | 0.50 |
| 750K | 6.49 | 5.39 | 0.81 | 0.50 |
| 1000K | 5.74 | 5.29 | 0.81 | 0.52 |
| 1500K | 5.01 | 5.20 | 0.82 | 0.52 |
| 1980K | 4.59 | 5.25 | 0.82 | 0.52 |

## B.3 Training Compute Comparison

Finally, in Table 10 we compare the compute of our models with StyleGAN2 and BigGAN-deep, and show we can obtain better FIDs with a similar compute budget. For BigGAN-deep, Brock et al. [8] do not explicitly describe the compute requirements for training their models, but rather provide rough estimates in terms of days on a Google TPUv3 pod [26]. We convert their TPU-v3 estimates to V100 days according to 2 TPU-v3 day = 1 V100 day. For StyleGAN2, we use the reported throughput of 25M images over 32 days 13 hour on one V100 for config-f [50]. We note that our classifier training is relatively lightweight compared to training the generative model.

Table 10: Training compute requirements for our diffusion models compared to StyleGAN2 and BigGAN-deep. Training iterations for each diffusion model are mentioned in parenthesis. Compute is measured in V100-days. [†]ImageNet 256×256 classifier with 150K iterations (instead of 500K). [‡]ImageNet 64×64 classifier with batch size 256 (instead of 1024). *ImageNet 128×128 classifier with batch size 256 (instead of 1024).

| Model | Generator Compute | Classifier Compute | Total Compute | FID | sFID | Precision | Recall |
|---|---|---|---|---|---|---|---|
| **LSUN Horse 256×256** | | | | | | | |
| StyleGAN2 [34] | | | 130 | 3.84 | 6.46 | 0.63 | 0.48 |
| ADM (250K) | 116 | - | **116** | 2.95 | **5.94** | 0.69 | **0.55** |
| ADM (dropout, 250K) | 116 | - | **116** | 2.57 | 6.81 | **0.71** | **0.55** |
| **LSUN Cat 256×256** | | | | | | | |
| StyleGAN2 [34] | | | 115 | 7.25 | **6.33** | 0.58 | 0.43 |
| ADM (dropout, 200K) | 92 | - | **92** | 5.57 | 6.69 | **0.63** | **0.52** |
| **ImageNet 128×128** | | | | | | | |
| BigGAN-deep [8] | | | 64-128 | 6.02 | 7.18 | **0.86** | 0.35 |
| ADM-G (4360K) | 521 | 9 | 530 | **3.09** | **5.59** | 0.82 | **0.54** |
| ADM-G (450K) | 54 | 9 | **63** | 5.67 | 6.19 | 0.82 | 0.49 |
| **ImageNet 256×256** | | | | | | | |
| BigGAN-deep [8] | | | 128-256 | 6.95 | 7.36 | **0.87** | 0.28 |
| ADM-G (1980K) | 916 | 46 | 962 | 4.59 | **5.25** | 0.82 | 0.52 |
| ADM-G (750K) | 347 | 46 | 393 | 6.49 | 5.39 | 0.81 | 0.50 |
| ADM-G (750K) | 347 | 14[†] | 361 | 6.68 | 5.34 | 0.81 | 0.51 |
| ADM-G (540K), ADM-U (500K) | 329 | 30 | 359 | **3.85** | 5.86 | 0.84 | 0.53 |
| ADM-G (540K), ADM-U (150K) | 219 | 30 | 249 | 4.15 | 6.14 | 0.82 | **0.54** |
| ADM-G (200K), ADM-U (150K) | 110 | 10[‡] | **126** | 4.93 | 5.82 | 0.82 | 0.52 |
| **ImageNet 512×512** | | | | | | | |
| BigGAN-deep [8] | | | 256-512 | 8.43 | 8.13 | **0.88** | 0.29 |
| ADM-G (4360K), ADM-U (1050K) | 1878 | 36 | 1914 | **3.85** | **5.86** | 0.84 | **0.53** |
| ADM-G (500K), ADM-U (100K) | 189 | 9* | **198** | 7.59 | 6.84 | 0.84 | **0.53** |

# C Detailed Formulation of DDPM

Here, we provide a detailed review of the formulation of Gaussian diffusion models from Ho et al. [31]. We start by defining our data distribution $x_0 \sim q(x_0)$ and a Markovian noising process $q$ which gradually adds noise to the data to produce noised samples $x_1$ through $x_T$. In particular, each step of the noising process adds Gaussian noise according to some variance schedule given by $\beta_t$:

$$q(x_t|x_{t-1}) := \mathcal{N}(x_t; \sqrt{1 - \beta_t}x_{t-1}, \beta_t\mathbf{I}) \tag{1}$$

Ho et al. [31] note that we need not apply $q$ repeatedly to sample from $x_t \sim q(x_t|x_0)$. Instead, $q(x_t|x_0)$ can be expressed as a Gaussian distribution. With $\alpha_t := 1 - \beta_t$ and $\bar{\alpha}_t := \prod_{s=0}^{t} \alpha_s$

$$q(x_t|x_0) = \mathcal{N}(x_t; \sqrt{\bar{\alpha}_t}x_0, (1 - \bar{\alpha}_t)\mathbf{I}) \tag{2}$$

$$= \sqrt{\bar{\alpha}_t}x_0 + \epsilon\sqrt{1 - \bar{\alpha}_t}, \ \epsilon \sim \mathcal{N}(0, \mathbf{I}) \tag{3}$$

Here, $1 - \bar{\alpha}_t$ tells us the variance of the noise for an arbitrary timestep, and we could equivalently use this to define the noise schedule instead of $\beta_t$.

Using Bayes theorem, one finds that the posterior $q(x_{t-1}|x_t, x_0)$ is also a Gaussian with mean $\tilde{\mu}_t(x_t, x_0)$ and variance $\tilde{\beta}_t$ defined as follows:

$$\tilde{\mu}_t(x_t, x_0) := \frac{\sqrt{\bar{\alpha}_{t-1}}\beta_t}{1 - \bar{\alpha}_t}x_0 + \frac{\sqrt{\alpha_t}(1 - \bar{\alpha}_{t-1})}{1 - \bar{\alpha}_t}x_t \tag{4}$$

$$\tilde{\beta}_t := \frac{1 - \bar{\alpha}_{t-1}}{1 - \bar{\alpha}_t}\beta_t \tag{5}$$

$$q(x_{t-1}|x_t, x_0) = \mathcal{N}(x_{t-1}; \tilde{\mu}(x_t, x_0), \tilde{\beta}_t\mathbf{I}) \tag{6}$$

If we wish to sample from the data distribution $q(x_0)$, we can first sample from $q(x_T)$ and then sample reverse steps $q(x_{t-1}|x_t)$ until we reach $x_0$. Under reasonable settings for $\beta_t$ and $T$, the distribution $q(x_T)$ is nearly an isotropic Gaussian distribution, so sampling $x_T$ is trivial. All that is left is to approximate $q(x_{t-1}|x_t)$ using a neural network, since it cannot be computed exactly when the data distribution is unknown. To this end, Sohl-Dickstein et al. [63] note that $q(x_{t-1}|x_t)$ approaches a diagonal Gaussian distribution as $T \to \infty$ and correspondingly $\beta_t \to 0$, so it is sufficient to train a neural network to predict a mean $\mu_\theta$ and a diagonal covariance matrix $\Sigma_\theta$:

$$p_\theta(x_{t-1}|x_t) := \mathcal{N}(x_{t-1}; \mu_\theta(x_t, t), \Sigma_\theta(x_t, t)) \tag{7}$$

To train this model such that $p(x_0)$ learns the true data distribution $q(x_0)$, we can optimize the following variational lower-bound $L_{\text{vlb}}$ for $p_\theta(x_0)$:

$$L_{\text{vlb}} := L_0 + L_1 + ... + L_{T-1} + L_T \tag{8}$$

$$L_0 := -\log p_\theta(x_0|x_1) \tag{9}$$

$$L_{t-1} := D_{KL}(q(x_{t-1}|x_t, x_0) \,||\, p_\theta(x_{t-1}|x_t)) \tag{10}$$

$$L_T := D_{KL}(q(x_T|x_0) \,||\, p(x_T)) \tag{11}$$

While the above objective is well-justified, Ho et al. [31] found that a different objective produces better samples in practice. In particular, they do not directly parameterize $\mu_\theta(x_t, t)$ as a neural network, but instead train a model $\epsilon_\theta(x_t, t)$ to predict $\epsilon$ from Equation 3. This simplified objective is defined as follows:

$$L_{\text{simple}} := E_{t \sim [1,T], x_0 \sim q(x_0), \epsilon \sim \mathcal{N}(0, \mathbf{I})}[||\epsilon - \epsilon_\theta(x_t, t)||^2] \tag{12}$$

During sampling, we can use substitution to derive $\mu_\theta(x_t, t)$ from $\epsilon_\theta(x_t, t)$:

$$\mu_\theta(x_t, t) = \frac{1}{\sqrt{\alpha_t}}\left(x_t - \frac{1 - \alpha_t}{\sqrt{1 - \bar{\alpha}_t}}\epsilon_\theta(x_t, t)\right) \tag{13}$$

Note that $L_{\text{simple}}$ does not provide any learning signal for $\Sigma_\theta(x_t, t)$. Ho et al. [31] find that instead of learning $\Sigma_\theta(x_t, t)$, they can fix it to a constant, choosing either $\beta_t\mathbf{I}$ or $\tilde{\beta}_t\mathbf{I}$. These values correspond to upper and lower bounds for the true reverse step variance.

# D  Derivations for Classifier Guidance

## D.1  Conditional Diffusion Process

We start by showing that conditional sampling can be achieved with a transition operator proportional to $p_\theta(x_t|x_{t+1})p_\phi(y|x_t)$, where $p_\theta(x_t|x_{t+1})$ approximates $q(x_t|x_{t+1})$ and $p_\phi(y|x_t)$ approximates the label distribution for a noised sample $x_t$.

We start by defining a conditional Markovian noising process $\hat{q}$ similar to $q$, and assume that $\hat{q}(y|x_0)$ is a known and readily available label distribution for each sample.

$$\hat{q}(x_0) := q(x_0) \tag{14}$$

$$\hat{q}(y|x_0) := \text{Known labels per sample} \tag{15}$$

$$\hat{q}(x_{t+1}|x_t, y) := q(x_{t+1}|x_t) \tag{16}$$

$$\hat{q}(x_{1:T}|x_0, y) := \prod_{t=1}^{T} \hat{q}(x_t|x_{t-1}, y) \tag{17}$$

While we defined the noising process $\hat{q}$ conditioned on $y$, we can prove that $\hat{q}$ behaves exactly like $q$ when not conditioned on $y$. Along these lines, we first derive the unconditional noising operator $\hat{q}(x_{t+1}|x_t)$:

$$\hat{q}(x_{t+1}|x_t) = \int_y \hat{q}(x_{t+1}, y|x_t)\,dy \tag{18}$$

$$= \int_y \hat{q}(x_{t+1}|x_t, y)\hat{q}(y|x_t)\,dy \tag{19}$$

$$= \int_y q(x_{t+1}|x_t)\hat{q}(y|x_t)\,dy \tag{20}$$

$$= q(x_{t+1}|x_t) \int_y \hat{q}(y|x_t)\,dy \tag{21}$$

$$= q(x_{t+1}|x_t) \tag{22}$$

$$= \hat{q}(x_{t+1}|x_t, y) \tag{23}$$

Following similar logic, we find the joint distribution $\hat{q}(x_{1:T}|x_0)$:

$$\hat{q}(x_{1:T}|x_0) = \int_y \hat{q}(x_{1:T}, y|x_0)\,dy \tag{24}$$

$$= \int_y \hat{q}(y|x_0)\hat{q}(x_{1:T}|x_0, y)\,dy \tag{25}$$

$$= \int_y \hat{q}(y|x_0) \prod_{t=1}^{T} \hat{q}(x_t|x_{t-1}, y)\,dy \tag{26}$$

$$= \int_y \hat{q}(y|x_0) \prod_{t=1}^{T} q(x_t|x_{t-1})\,dy \tag{27}$$

$$= \prod_{t=1}^{T} q(x_t|x_{t-1}) \int_y \hat{q}(y|x_0)\,dy \tag{28}$$

$$= \prod_{t=1}^{T} q(x_t|x_{t-1}) \tag{29}$$

$$= q(x_{1:T}|x_0) \tag{30}$$

Using Equation 30, we can now derive $\hat{q}(x_t)$:

$$\hat{q}(x_t) = \int_{x_{0:t-1}} \hat{q}(x_0, ..., x_t)\, dx_{0:t-1} \tag{31}$$

$$= \int_{x_{0:t-1}} \hat{q}(x_0)\hat{q}(x_1, ..., x_t|x_0)\, dx_{0:t-1} \tag{32}$$

$$= \int_{x_{0:t-1}} q(x_0)q(x_1, ..., x_t|x_0)\, dx_{0:t-1} \tag{33}$$

$$= \int_{x_{0:t-1}} q(x_0, ..., x_t)\, dx_{0:t-1} \tag{34}$$

$$= q(x_t) \tag{35}$$

$$\tag{36}$$

Using the identities $\hat{q}(x_t) = q(x_t)$ and $\hat{q}(x_{t+1}|x_t) = q(x_{t+1}|x_t)$, it is trivial to show via Bayes rule that the unconditional reverse process $\hat{q}(x_t|x_{t+1}) = q(x_t|x_{t+1})$.

One observation about $\hat{q}$ is that it gives rise to a noisy classification function, $\hat{q}(y|x_t)$. We can show that this classification distribution does not depend on $x_{t+1}$ (a noisier version of $x_t$), a fact which we will later use:

$$\hat{q}(y|x_t, x_{t+1}) = \hat{q}(x_{t+1}|x_t, y)\frac{\hat{q}(y|x_t)}{\hat{q}(x_{t+1}|x_t)} \tag{37}$$

$$= \hat{q}(x_{t+1}|x_t)\frac{\hat{q}(y|x_t)}{\hat{q}(x_{t+1}|x_t)} \tag{38}$$

$$= \hat{q}(y|x_t) \tag{39}$$

$$\tag{40}$$

We can now derive the conditional reverse process:

$$\hat{q}(x_t|x_{t+1}, y) = \frac{\hat{q}(x_t, x_{t+1}, y)}{\hat{q}(x_{t+1}, y)} \tag{41}$$

$$= \frac{\hat{q}(x_t, x_{t+1}, y)}{\hat{q}(y|x_{t+1})\hat{q}(x_{t+1})} \tag{42}$$

$$= \frac{\hat{q}(x_t|x_{t+1})\hat{q}(y|x_t, x_{t+1})\hat{q}(x_{t+1})}{\hat{q}(y|x_{t+1})\hat{q}(x_{t+1})} \tag{43}$$

$$= \frac{\hat{q}(x_t|x_{t+1})\hat{q}(y|x_t, x_{t+1})}{\hat{q}(y|x_{t+1})} \tag{44}$$

$$= \frac{\hat{q}(x_t|x_{t+1})\hat{q}(y|x_t)}{\hat{q}(y|x_{t+1})} \tag{45}$$

$$= \frac{q(x_t|x_{t+1})\hat{q}(y|x_t)}{\hat{q}(y|x_{t+1})} \tag{46}$$

$$\tag{47}$$

The $\hat{q}(y|x_{t+1})$ term can be treated as a constant since it does not depend on $x_t$. We thus want to sample from the distribution $Zq(x_t|x_{t+1})\hat{q}(y|x_t)$ where $Z$ is a normalizing constant. We already have a neural network approximation of $q(x_t|x_{t+1})$, called $p_\theta(x_t|x_{t+1})$, so all that is left is an approximation of $\hat{q}(y|x_t)$. This can be obtained by training a classifier $p_\phi(y|x_t)$ on noised images $x_t$ derived by sampling from $q(x_t)$.

## D.2 Deriving Algorithm 1: Conditional Sampling for DDPM

We showed in the previous section that to condition a diffusion process on a label $y$, it suffices to sample each transition[2] according to

$$p_{\theta,\phi}(x_t|x_{t+1}, y) = Z p_\theta(x_t|x_{t+1}) p_\phi(y|x_t) \tag{48}$$

where $Z$ is a normalizing constant. It is typically intractable to sample from this distribution exactly, but Sohl-Dickstein et al. [63] show that it can be approximated as a perturbed Gaussian distribution. Here, we review this derivation.

Recall that our diffusion model predicts the previous timestep $x_t$ from timestep $x_{t+1}$ using a Gaussian distribution:

$$p_\theta(x_t|x_{t+1}) = \mathcal{N}(\mu, \Sigma) \tag{49}$$

$$\log p_\theta(x_t|x_{t+1}) = -\frac{1}{2}(x_t - \mu)^T \Sigma^{-1}(x_t - \mu) + C \tag{50}$$

We can assume that $\log_\phi p(y|x_t)$ has low curvature compared to $\Sigma^{-1}$. This assumption is reasonable in the limit of infinite diffusion steps, where $||\Sigma|| \to 0$. In this case, we can approximate $\log p_\phi(y|x_t)$ using a Taylor expansion around $x_t = \mu$ as

$$\log p_\phi(y|x_t) \approx \log p_\phi(y|x_t)|_{x_t=\mu} + (x_t - \mu) \nabla_{x_t} \log p_\phi(y|x_t)|_{x_t=\mu} \tag{51}$$

$$= (x_t - \mu)g + C_1 \tag{52}$$

Here, $g = \nabla_{x_t} \log p_\phi(y|x_t)|_{x_t=\mu}$, and $C_1$ is a constant. This gives

$$\log(p_\theta(x_t|x_{t+1}) p_\phi(y|x_t)) \approx -\frac{1}{2}(x_t - \mu)^T \Sigma^{-1}(x_t - \mu) + (x_t - \mu)g + C_2 \tag{53}$$

$$= -\frac{1}{2}(x_t - \mu - \Sigma g)^T \Sigma^{-1}(x_t - \mu - \Sigma g) + \frac{1}{2}g^T \Sigma g + C_2 \tag{54}$$

$$= -\frac{1}{2}(x_t - \mu - \Sigma g)^T \Sigma^{-1}(x_t - \mu - \Sigma g) + C_3 \tag{55}$$

$$= \log p(z) + C_4, z \sim \mathcal{N}(\mu + \Sigma g, \Sigma) \tag{56}$$

We can safely ignore the constant term $C_4$, since it corresponds to the normalizing coefficient $Z$ in Equation 48. We have thus found that the conditional transition operator can be approximated by a Gaussian similar to the unconditional transition operator, but with its mean shifted by $\Sigma g$.

## D.3 Deriving Algorithm 2: Conditional Sampling for DDIM

The above derivation for conditional sampling is only valid for the stochastic diffusion sampling process, and cannot be applied to deterministic sampling methods like DDIM [64]. To this end, we use a score-based conditioning trick adapted from Song et al. [67], which leverages the connection between diffusion models and score matching [66]. In particular, if we have a model $\epsilon_\theta(x_t)$ that predicts the noise added to a sample, then this can be used to derive a score function:

$$\nabla_{x_t} \log p_\theta(x_t) = -\frac{1}{\sqrt{1 - \bar{\alpha}_t}} \epsilon_\theta(x_t) \tag{57}$$

We can now substitute this into the score function for $p(x_t)p(y|x_t)$:

$$\nabla_{x_t} \log(p_\theta(x_t) p_\phi(y|x_t)) = \nabla_{x_t} \log p_\theta(x_t) + \nabla_{x_t} \log p_\phi(y|x_t) \tag{58}$$

$$= -\frac{1}{\sqrt{1 - \bar{\alpha}_t}} \epsilon_\theta(x_t) + \nabla_{x_t} \log p_\phi(y|x_t) \tag{59}$$

Finally, we can define a new epsilon prediction $\hat{\epsilon}(x_t)$ which corresponds to the score of the joint distribution:

$$\hat{\epsilon}(x_t) := \epsilon_\theta(x_t) - \sqrt{1 - \bar{\alpha}_t} \nabla_{x_t} \log p_\phi(y|x_t) \tag{60}$$

We can then use the exact same sampling procedure as used for regular DDIM, but with the modified noise predictions $\hat{\epsilon}_\theta(x_t)$ instead of $\epsilon_\theta(x_t)$.

---

[2]We must also sample $x_T$ conditioned on $y$, but a noisy enough diffusion process causes $x_T$ to be nearly Gaussian even in the conditional case.

# E    Nearest Neighbors for Samples

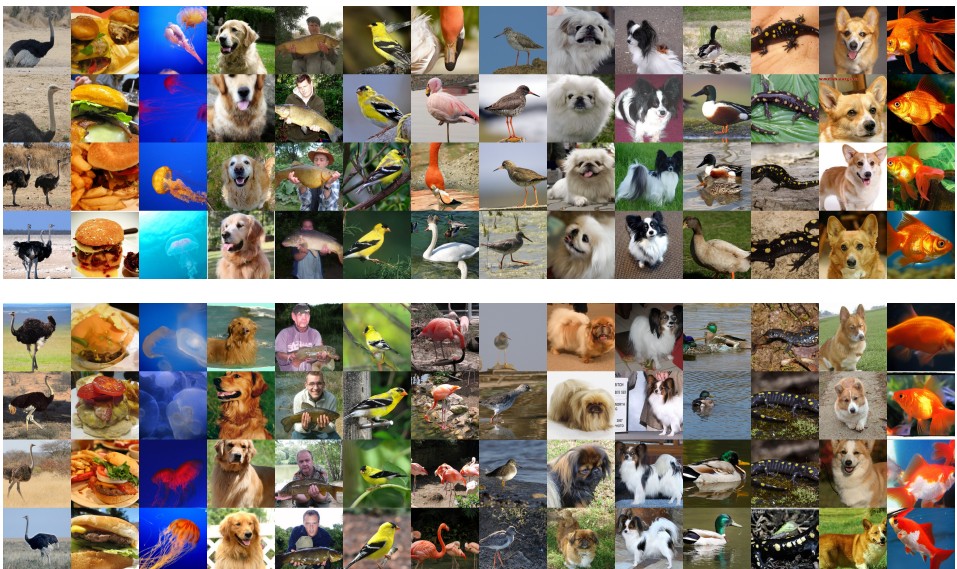

Figure 7: Nearest neighbors for samples from a classifier guided model on ImageNet 256×256. For each image, the top row is a sample, and the remaining rows are the top 3 nearest neighbors from the dataset. The top samples were generated with classifier scale 1 and 250 diffusion sampling steps (FID 4.59). The bottom samples were generated with classifier scale 2.5 and 25 DDIM steps (FID 5.44).

Our models achieve their best FID when using a classifier to reduce the diversity of the generations. One might fear that such a process could cause the model to recall existing images from the training dataset, especially as the classifier scale is increased. To test this, we looked at the nearest neighbors (in InceptionV3 [68] feature space) for a handful of samples. Figure 7 shows our results, revealing that the samples are indeed unique and not stored in the training set.

# F    Effect of Varying the Classifier Scale

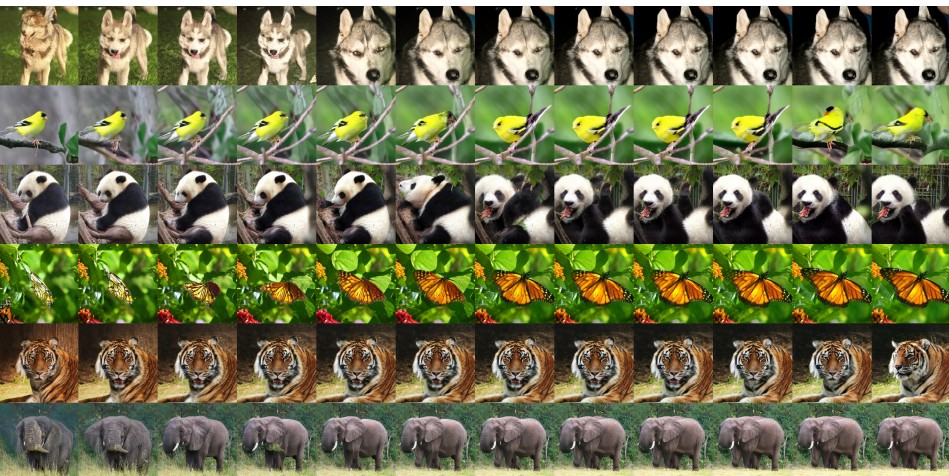

Figure 8: Samples when increasing the classifier scale from 0.0 (left) to 5.5 (right) for a class conditional ImageNet 256×256 model. Each row corresponds to a fixed noise seed. We observe that the classifier drastically changes some images, while leaving others relatively unaffected.

# G  LSUN Diversity Comparison

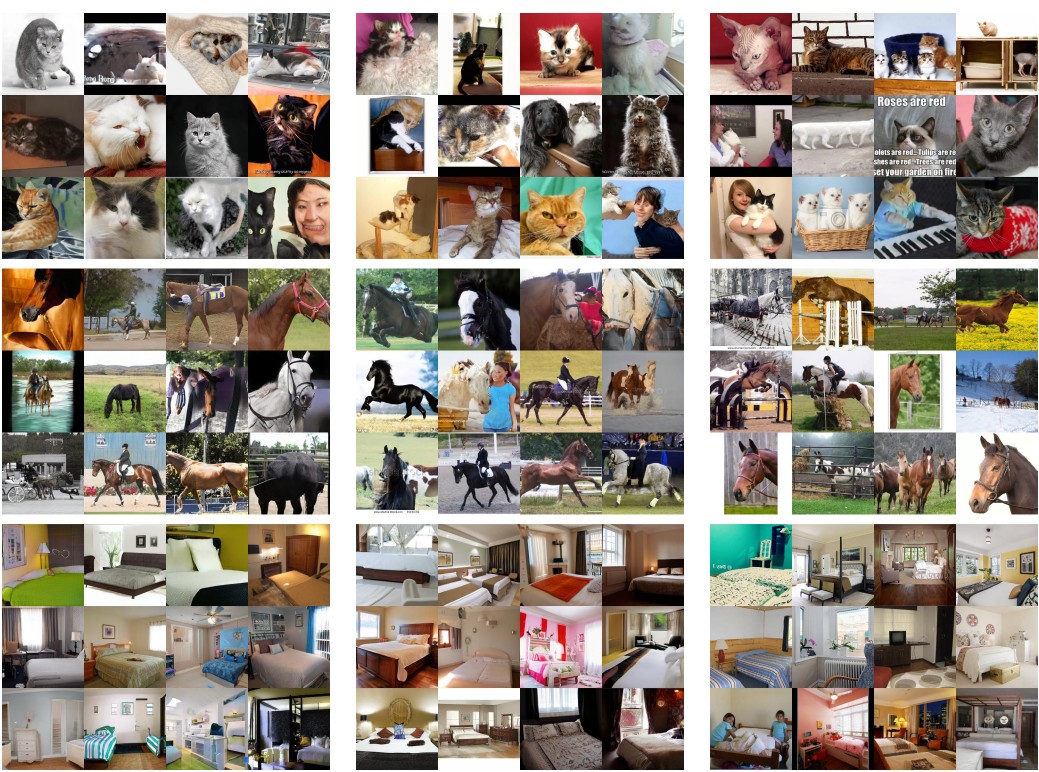

Figure 9: Samples from StyleGAN2 (or StyleGAN for bedrooms) with truncation 1.0 (left) vs samples from our diffusion models (middle) and samples from the training set (right).

# H   Interpolating Between Dataset Images Using DDIM

The DDIM [64] sampling process is deterministic given the initial noise $x_T$, thus giving rise to an implicit latent space. It corresponds to integrating an ODE in the forward direction, and we can run the process in reverse to get the latents that produce a given real image. Here, we experiment with encoding real images into this latent space and then interpolating between them.

Equation 13 for the generative pass in DDIM looks like

$$x_{t-1} - x_t = \sqrt{\bar{\alpha}_{t-1}} \left[ \left( \sqrt{1/\bar{\alpha}_t} - \sqrt{1/\bar{\alpha}_{t-1}} \right) x_t + \left( \sqrt{1/\bar{\alpha}_{t-1} - 1} - \sqrt{1/\bar{\alpha}_t - 1} \right) \epsilon_\theta(x_t) \right]$$

Thus, in the limit of small steps, we can expect the reversal of this ODE in the forward direction looks like

$$x_{t+1} - x_t = \sqrt{\bar{\alpha}_{t+1}} \left[ \left( \sqrt{1/\bar{\alpha}_t} - \sqrt{1/\bar{\alpha}_{t+1}} \right) x_t + \left( \sqrt{1/\bar{\alpha}_{t+1} - 1} - \sqrt{1/\bar{\alpha}_t - 1} \right) \epsilon_\theta(x_t) \right]$$

We found that this reverse ODE approximation allows us to obtain latents for real images with reasonable reconstructions, even with as few as 250 reverse steps. However, we noticed some noise artifacts when reversing all 250 steps, and find that reversing the first 249 steps gives much better reconstructions. To interpolate the latents, class embeddings, and classifier log probabilities, we use $cos(\theta)x_0 + sin(\theta)x_1$ where $\theta$ sweeps linearly from 0 to $\frac{\pi}{2}$.

Figures $10a$ through $10c$ show DDIM latent space interpolations on a class-conditional $256 \times 256$ model, while varying the classifier scale. The left and rightmost images are randomly selected ground truth dataset examples, and between them are reconstructed interpolations in DDIM latent space (including both endpoints). We see that the model with no guidance has almost perfect reconstructions due to its high recall, whereas raising the guidance scale to 2.5 only finds approximately similar reconstructions.

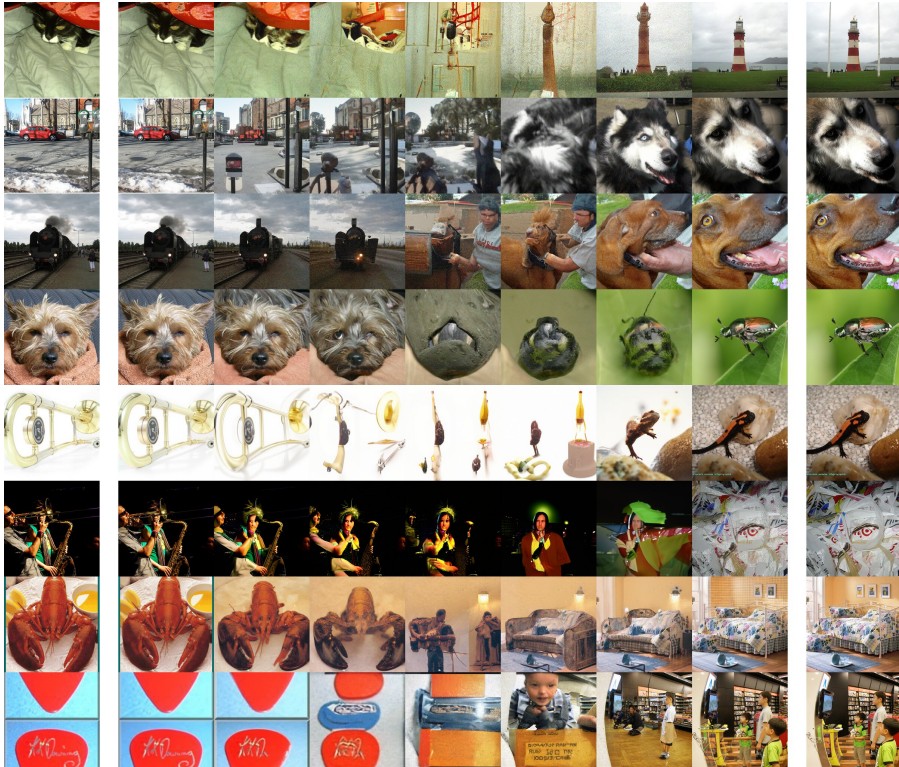

Figure 10a:  DDIM latent reconstructions and interpolations on real images with no classifier guidance.

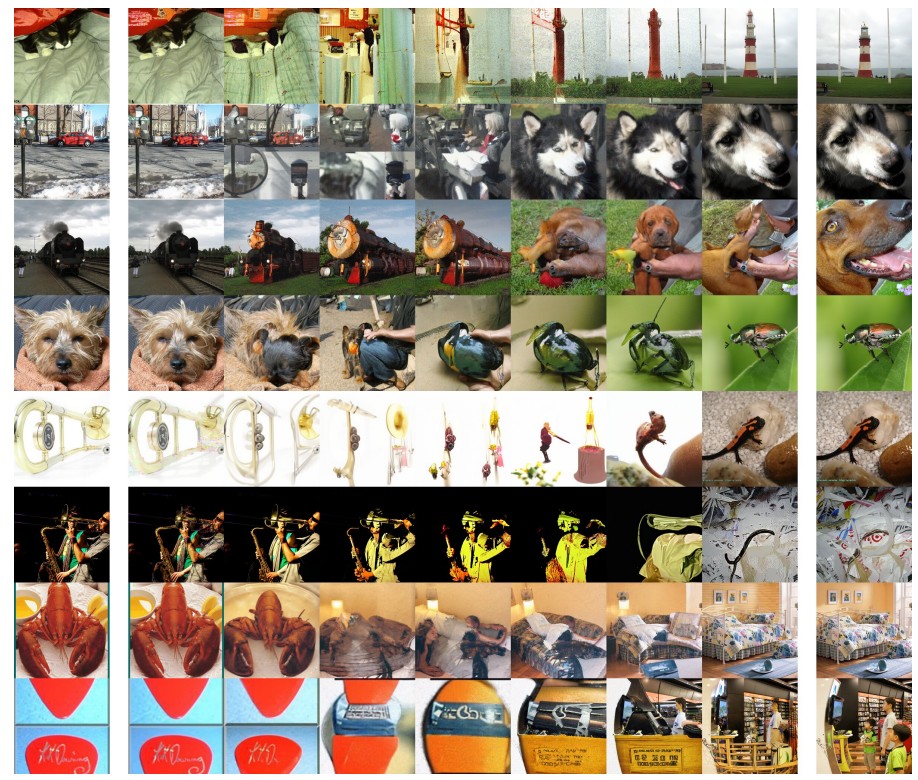

Figure 10b: DDIM latent reconstructions and interpolations on real images with classifier scale 1.0.

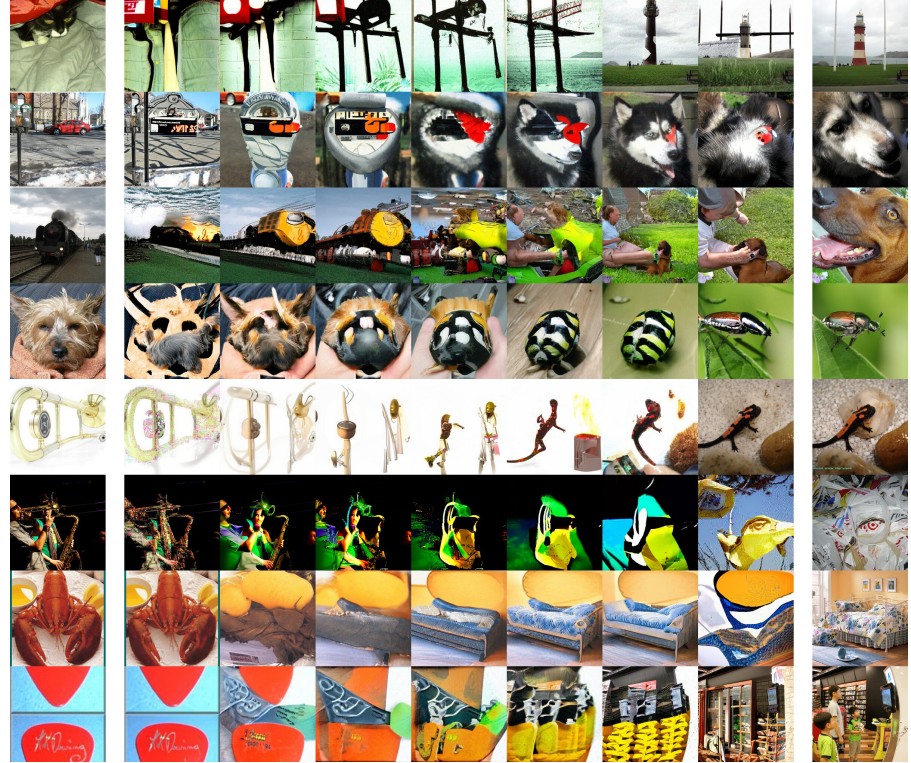

Figure 10c: DDIM latent reconstructions and interpolations on real images with classifier scale 2.5.

# I Reduced Temperature Sampling

We achieved our best ImageNet samples by reducing the diversity of our models using classifier guidance. For many classes of generative models, there is a much simpler way to reduce diversity: reducing the temperature [1]. The temperature parameter $\tau$ is typically setup so that $\tau = 1.0$ corresponds to standard sampling, and $\tau < 1.0$ focuses more on high-density samples. We experimented with two ways of implementing this for diffusion models: first, by scaling the Gaussian noise used for each transition by $\tau$, and second by dividing $\epsilon_\theta(x_t)$ by $\tau$. The latter implementation makes sense when thinking about $\epsilon$ as a re-scaled score function (see Appendix D.3), and scaling up the score function is similar to scaling up classifier gradients.

To measure how temperature scaling affects samples, we experimented with our ImageNet $128\times128$ model, evaluating FID, Precision, and Recall across different temperatures (Figure 11). We find that two techniques behave similarly, and neither technique provides any substantial improvement in our evaluation metrics. We also find that low temperatures have both low precision and low recall, indicating that the model is not focusing on modes of the real data distribution. Figure 12 highlights this effect, indicating that reducing temperature produces blurry, smooth images.

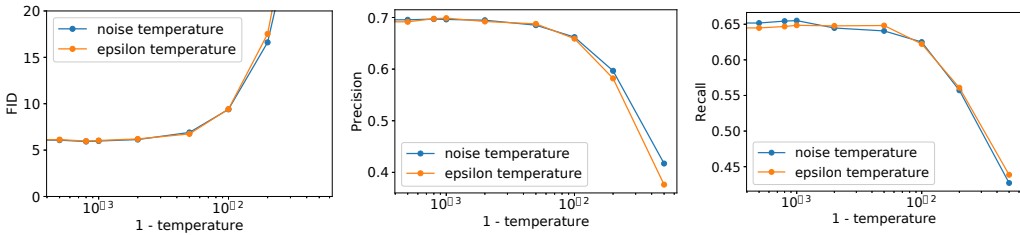

Figure 11: The effect of changing temperature for an ImageNet $128\times128$ model.

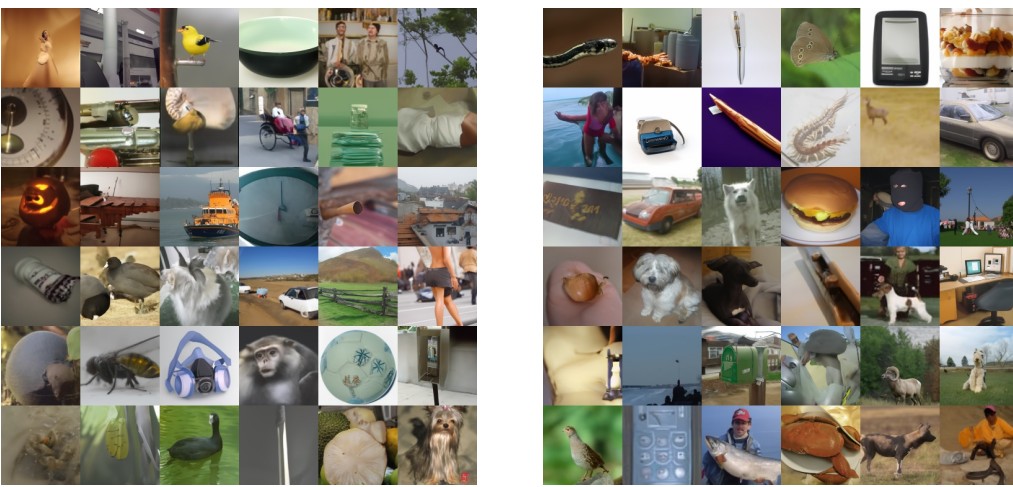

Figure 12: Samples at temperature 0.98 with epsilon scaling (left) and noise scaling (right).

## J  Sample Quality Metrics

Inception Score (IS) was proposed by Salimans et al. [61], and it measures how well a model captures the full ImageNet class distribution while still producing individual samples that are convincing examples of a single class. One drawback of this metric is that it does not reward covering the whole distribution or capturing diversity within a class, and models which memorize a small subset of the full dataset will still have high IS [6]. To better capture diversity than IS, Fréchet Inception Distance (FID) was proposed by Heusel et al. [29], who argued that it is more consistent with human judgement than Inception Score. FID provides a symmetric measure of the distance between two image distributions in the Inception-V3 [68] latent space. Recently, sFID was proposed by Nash et al. [48] as a version of FID that uses spatial features rather than the standard pooled features. They find that this metric better captures spatial relationships, rewarding image distributions with coherent high-level structure. Finally, Kynkäänniemi et al. [38] proposed Improved Precision and Recall metrics to separately measure sample fidelity as the fraction of model samples which fall into the data manifold (precision), and diversity as the fraction of data samples which fall into the sample manifold (recall).

We compute precision and recall with K=3, using 50K model samples and 10K reference samples from the training set. We compute all FIDs against the entire training set, using 50K samples from the model. For StyleGAN and StyleGAN2, we use the first 50K samples from the official release repositories. For BigGAN-deep, we sample from the officially released models [15] at truncation level 1.0 (or lower truncations when applicable). We report DCTransformer and VQ-VAE-2 evaluations from Nash et al. [48].

When computing FID, we follow the original FID implementation [29] and do not resize images prior to feeding them into the Inception graph. This may differ from unofficial FID implementations, and can cause a slight difference in FID values [51]. This should not affect relative comparisons against GANs, since we use our FID implementation to evaluate samples from all GAN models.

## K  Hyperparameters

When choosing optimal classifier scales for our sampler, we swept over $[0.5, 1, 2]$ for ImageNet 128×128 and ImageNet 256×256, and $[1, 2, 3, 3.5, 4, 4.5, 5]$ for ImageNet 512×512. For DDIM, we swept over values $[0.5, 0.75, 1.0, 1.25, 2]$ for ImageNet 128×128, $[0.5, 1, 1.5, 2, 2.5, 3, 3.5]$ for ImageNet 256×256, and $[3, 4, 5, 6, 7, 9, 11]$ for ImageNet 512×512.

Hyperparameters for training the diffusion and classification models are in Table 11 and Table 12 respectively. Hyperparameters for guided sampling are in Table 14. Hyperparameters used to train upsampling models are in Table 13. We train all of our models using Adam [35] or AdamW [41] with $\beta_1 = 0.9$ and $\beta_2 = 0.999$. We train in 16-bit precision using loss-scaling [44], but maintain 32-bit weights, EMA, and optimizer state. We use an EMA rate of 0.9999 for all experiments. We use PyTorch [52], and train on NVIDIA Tesla V100s.

For all architecture ablations, we train with batch size 256, and sample using 250 sampling steps. For our attention heads ablations, we use 128 base channels, 2 residual blocks per resolution, multi-resolution attention, and BigGAN up/downsampling, and we train the models for 700K iterations. By default, all of our experiments use adaptive group normalization, except when explicitly ablating for it.

When sampling with 1000 timesteps, we use the same noise schedule as for training. On ImageNet, we use the uniform stride from Nichol and Dhariwal [49] for 250 step samples and the slightly different uniform stride from Song et al. [64] for 25 step DDIM.

Table 11: Hyperparameters for diffusion models. *We used 200K iterations for LSUN cat, 250K for LSUN horse, and 500K for LSUN bedroom.

| | LSUN | ImageNet 64 | ImageNet 128 | ImageNet 256 | ImageNet 512 |
|---|---|---|---|---|---|
| Diffusion steps | 1000 | 1000 | 1000 | 1000 | 1000 |
| Noise Schedule | linear | cosine | linear | linear | linear |
| Model size | 552M | 296M | 422M | 554M | 559M |
| Channels | 256 | 192 | 256 | 256 | 256 |
| Depth | 2 | 3 | 2 | 2 | 2 |
| Channels multiple | 1,1,2,2,4,4 | 1,2,3,4 | 1,1,2,3,4 | 1,1,2,2,4,4 | 0.5,1,1,2,2,4,4 |
| Heads | | | 4 | | |
| Heads Channels | 64 | 64 | | 64 | 64 |
| Attention resolution | 32,16,8 | 32,16,8 | 32,16,8 | 32,16,8 | 32,16,8 |
| BigGAN up/downsample | ✓ | ✓ | ✓ | ✓ | ✓ |
| Dropout | 0.1 | 0.1 | 0.0 | 0.0 | 0.0 |
| Batch size | 256 | 2048 | 256 | 256 | 256 |
| Iterations | varies* | 540K | 4360K | 1980K | 1940K |
| Learning Rate | 1e-4 | 3e-4 | 1e-4 | 1e-4 | 1e-4 |

Table 12: Hyperparameters for classification models. *For our ImageNet 128 → 512 upsamples, we use a different classifier, trained with batch size 1024 and learning rate 6e-5.

| | ImageNet 64 | ImageNet 128 | ImageNet 256 | ImageNet 512 |
|---|---|---|---|---|
| Diffusion steps | 1000 | 1000 | 1000 | 1000 |
| Noise Schedule | cosine | linear | linear | linear |
| Model size | 65M | 43M | 54M | 54M |
| Channels | 128 | 128 | 128 | 128 |
| Depth | 4 | 2 | 2 | 2 |
| Channels multiple | 1,2,3,4 | 1,1,2,3,4 | 1,1,2,2,4,4 | 0.5,1,1,2,2,4,4 |
| Heads Channels | 64 | 64 | 64 | 64 |
| Attention resolution | 32,16,8 | 32,16,8 | 32,16,8 | 32,16,8 |
| BigGAN up/downsample | ✓ | ✓ | ✓ | ✓ |
| Attention pooling | ✓ | ✓ | ✓ | ✓ |
| Weight decay | 0.2 | 0.05 | 0.05 | 0.05 |
| Batch size | 1024 | 256* | 256 | 256 |
| Iterations | 300K | 300K | 500K | 500K |
| Learning rate | 6e-4 | 3e-4* | 3e-4 | 3e-4 |

Table 13: Hyperparameters for upsampling diffusion models. *We chose this as an optimization, with the intuition that a lower-resolution path should be unnecessary for upsampling 128x128 images.

| | ImageNet 64 → 256 | ImageNet 128 → 512 |
|---|---|---|
| Diffusion steps | 1000 | 1000 |
| Noise Schedule | linear | linear |
| Model size | 312M | 309M |
| Channels | 192 | 192 |
| Depth | 2 | 2 |
| Channels multiple | 1,1,2,2,4,4 | 1,1,2,2,4,4* |
| Heads | 4 | |
| Heads Channels | | 64 |
| Attention resolution | 32,16,8 | 32,16,8 |
| BigGAN up/downsample | ✓ | ✓ |
| Dropout | 0.0 | 0.0 |
| Batch size | 256 | 256 |
| Iterations | 500K | 1050K |
| Learning Rate | 1e-4 | 1e-4 |

Table 14: Hyperparameters for classifier-guided sampling.

| | ImageNet 64 | ImageNet 128 | ImageNet 256 | ImageNet 512 |
|---|---|---|---|---|
| Gradient Scale (250 steps) | 1.0 | 0.5 | 1.0 | 4.0 |
| Gradient Scale (DDIM, 25 steps) | - | 1.25 | 2.5 | 9.0 |

# L  Using Fewer Sampling Steps on LSUN

We initially found that our LSUN models achieved much better results when sampling with 1000 steps rather than 250 steps, contrary to previous results from Nichol and Dhariwal [49]. To address this, we conducted a sweep over sampling-time noise schedules, finding that an improved schedule can largely close the gap. We swept over schedules on LSUN bedrooms, and selected the schedule with the best FID for use on the other two datasets. Table 15 details the findings of this sweep, and Table 16 applies this schedule to three LSUN datasets.

While sweeping over sampling schedules is not as expensive as re-training models from scratch, it does require a significant amount of sampling compute. As a result, we did not conduct an exhaustive sweep, and superior schedules are likely to exist.

Table 15: Results of sweeping over 250 step sampling schedules on LSUN bedrooms. The schedule is expressed as a sequence of five integers, where each integer is the number of steps allocated to one fifth of the diffusion process. The first integer corresponding to $t \in [0, 199]$ and the last to $t \in [T - 200, T - 1]$. Thus, $50, 50, 50, 50, 50$ is a uniform schedule, and $250, 0, 0, 0, 0$ is a schedule where all timesteps are spent near $t = 0$.

| Schedule | FID |
|---|---|
| $50, 50, 50, 50, 50$ | 2.31 |
| $70, 60, 50, 40, 30$ | 2.17 |
| $90, 50, 40, 40, 30$ | 2.10 |
| $90, 60, 50, 30, 20$ | 2.09 |
| $80, 60, 50, 30, 30$ | 2.09 |
| $90, 50, 50, 30, 30$ | 2.07 |
| $100, 50, 40, 30, 30$ | 2.03 |
| $90, 60, 60, 20, 20$ | **2.02** |

Table 16: Evaluations on LSUN bedrooms, horses, and cats using different sampling schedules. We find that the sweep schedule produces better results than the uniform 250 step schedule on all three datasets, and mostly bridges the gap to the 1000 step schedule.

| Schedule | FID | sFID | Prec | Rec |
|---|---|---|---|---|
| **LSUN Bedrooms 256×256** | | | | |
| 1000 steps | 1.90 | 5.59 | 0.66 | 0.51 |
| 250 steps (uniform) | 2.31 | 6.12 | 0.65 | 0.50 |
| 250 steps (sweep) | 2.02 | 6.12 | 0.67 | 0.50 |
| **LSUN Horses 256×256** | | | | |
| 1000 steps | 2.57 | 6.81 | 0.71 | 0.55 |
| 250 steps (uniform) | 3.45 | 7.55 | 0.68 | 0.56 |
| 250 steps (sweep) | 2.83 | 7.08 | 0.69 | 0.56 |
| **LSUN Cat 256×256** | | | | |
| 1000 steps | 5.57 | 6.69 | 0.63 | 0.52 |
| 250 steps (uniform) | 7.03 | 8.24 | 0.60 | 0.53 |
| 250 steps (sweep) | 5.94 | 7.43 | 0.62 | 0.52 |

## M    Samples from ImageNet 512×512

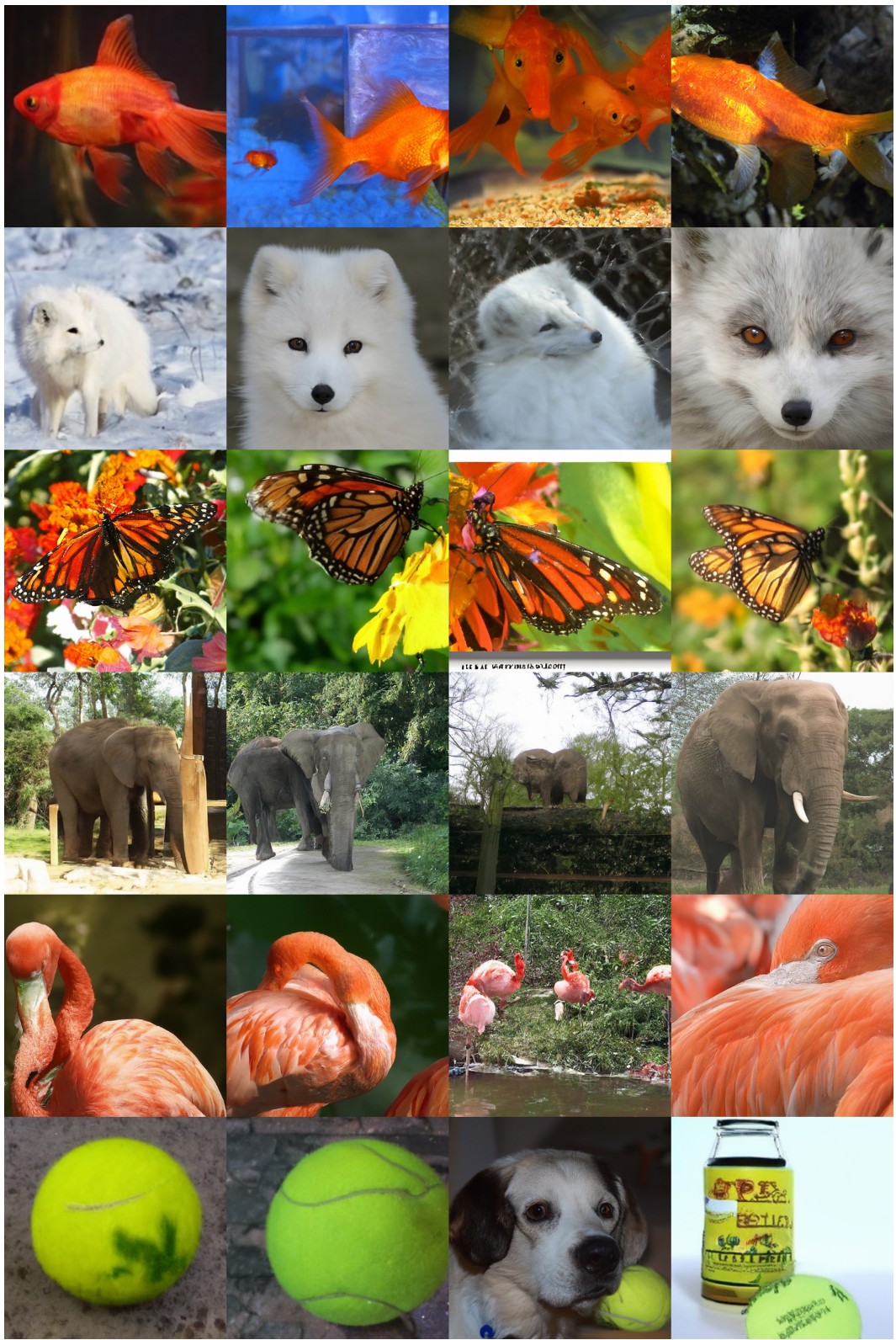

Figure 13: Samples from our best 512×512 model (FID: 3.85). Classes are 1: goldfish, 279: arctic fox, 323: monarch butterfly, 386: african elephant, 130: flamingo, 852: tennis ball.

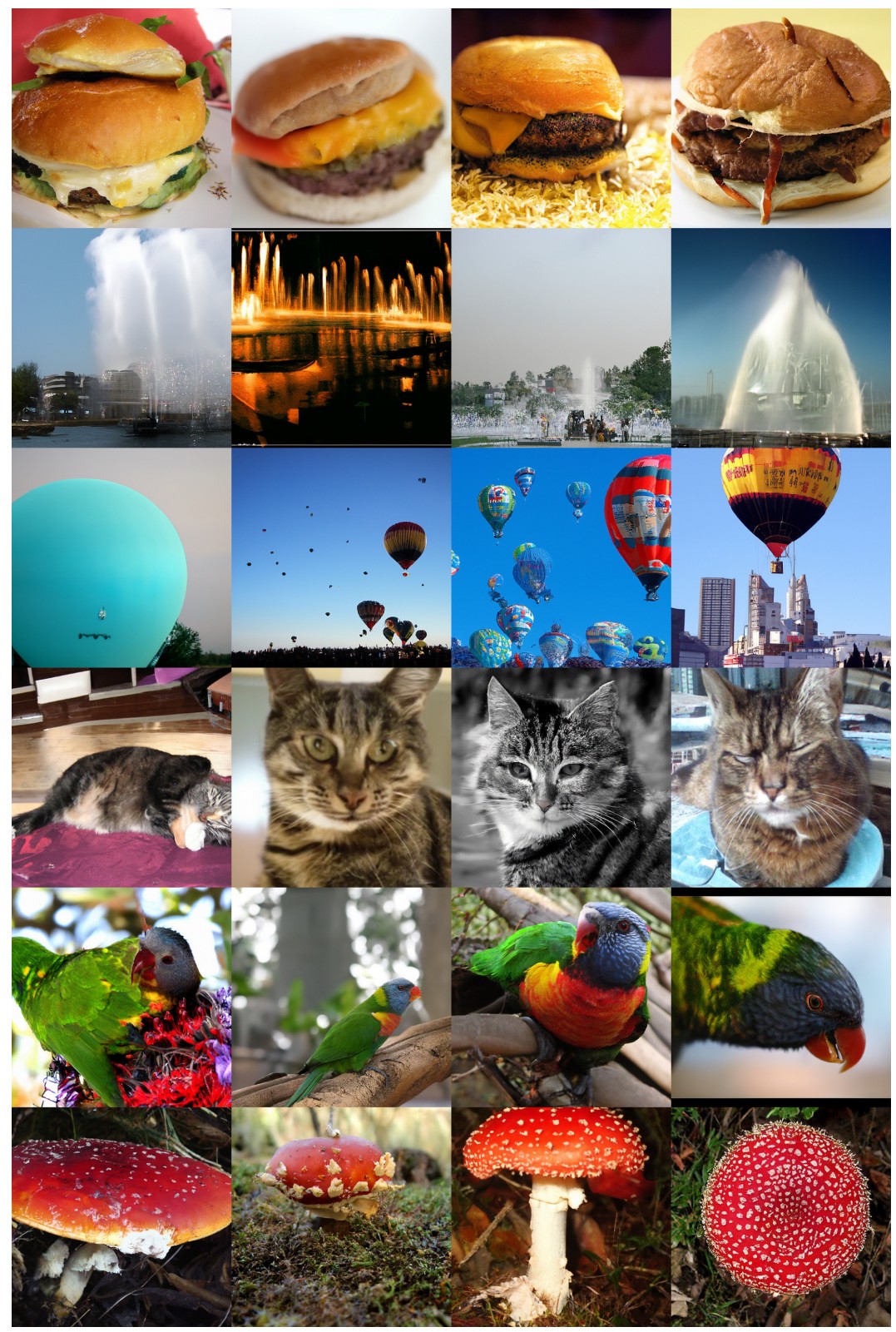

Figure 14: Samples from our best 512×512 model (FID: 3.85). Classes are 933: cheeseburger, 562: fountain, 417: balloon, 281: tabby cat, 90: lorikeet, 992: agaric.

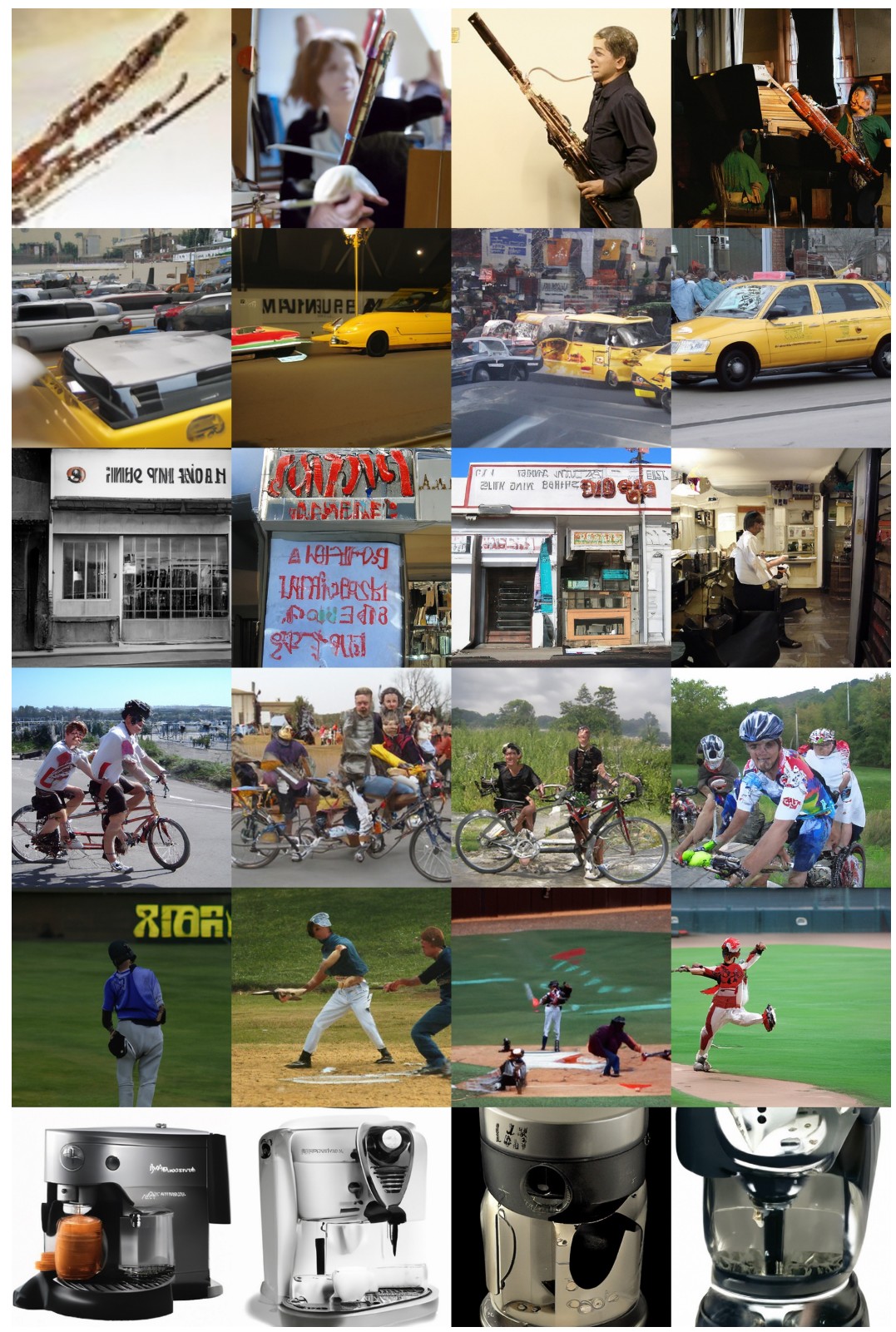

Figure 15: Difficult class samples from our best 512×512 model (FID: 3.85). Classes are 432: bassoon, 468: cab, 424: barbershop, 444: bicycle-built-for-two, 981: ballplayer, 550: espresso maker.

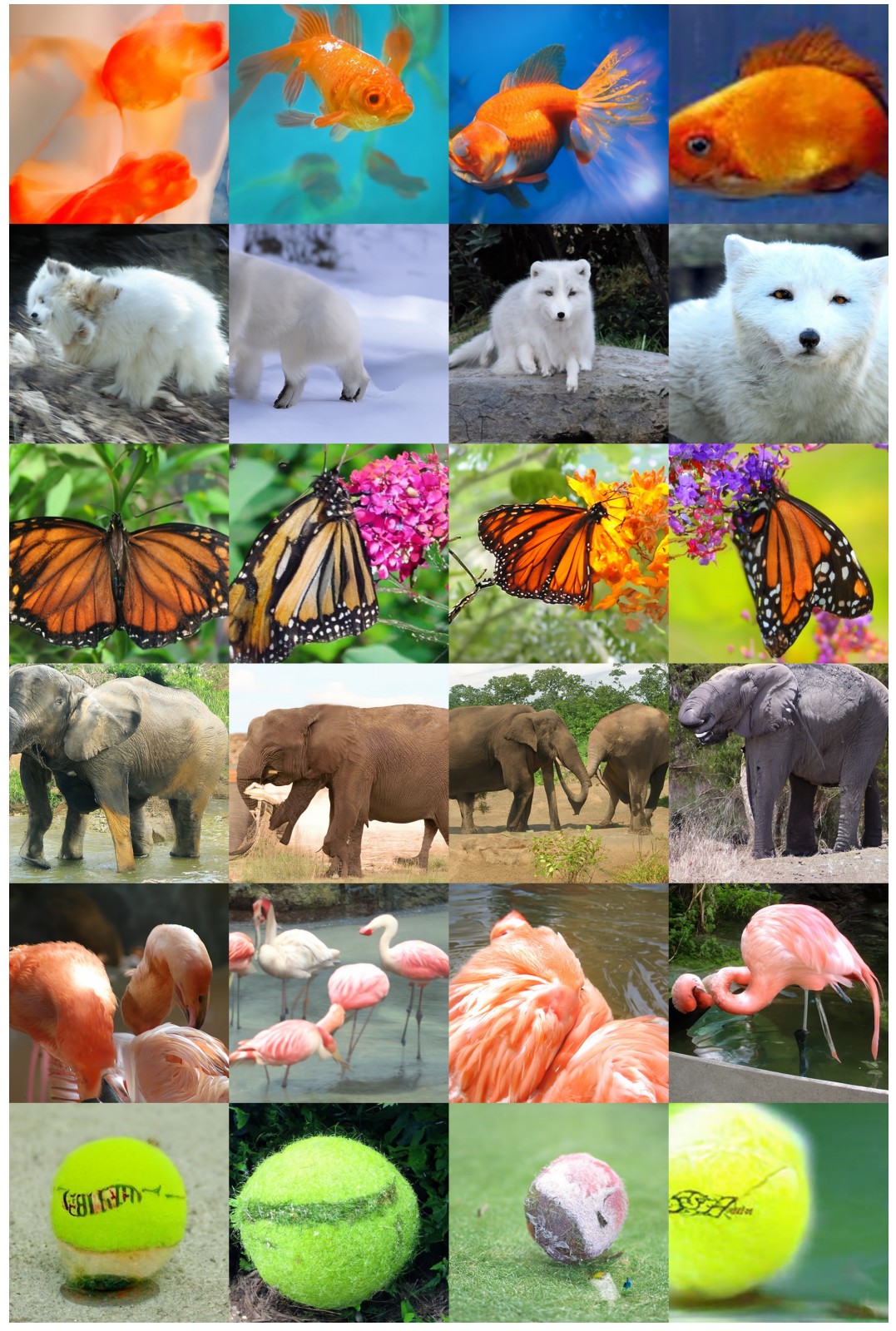

Figure 16: Samples from our guided 512×512 model using 250 steps with classifier scale 4.0 (FID 7.72). Classes are 1: goldfish, 279: arctic fox, 323: monarch butterfly, 386: african elephant, 130: flamingo, 852: tennis ball.

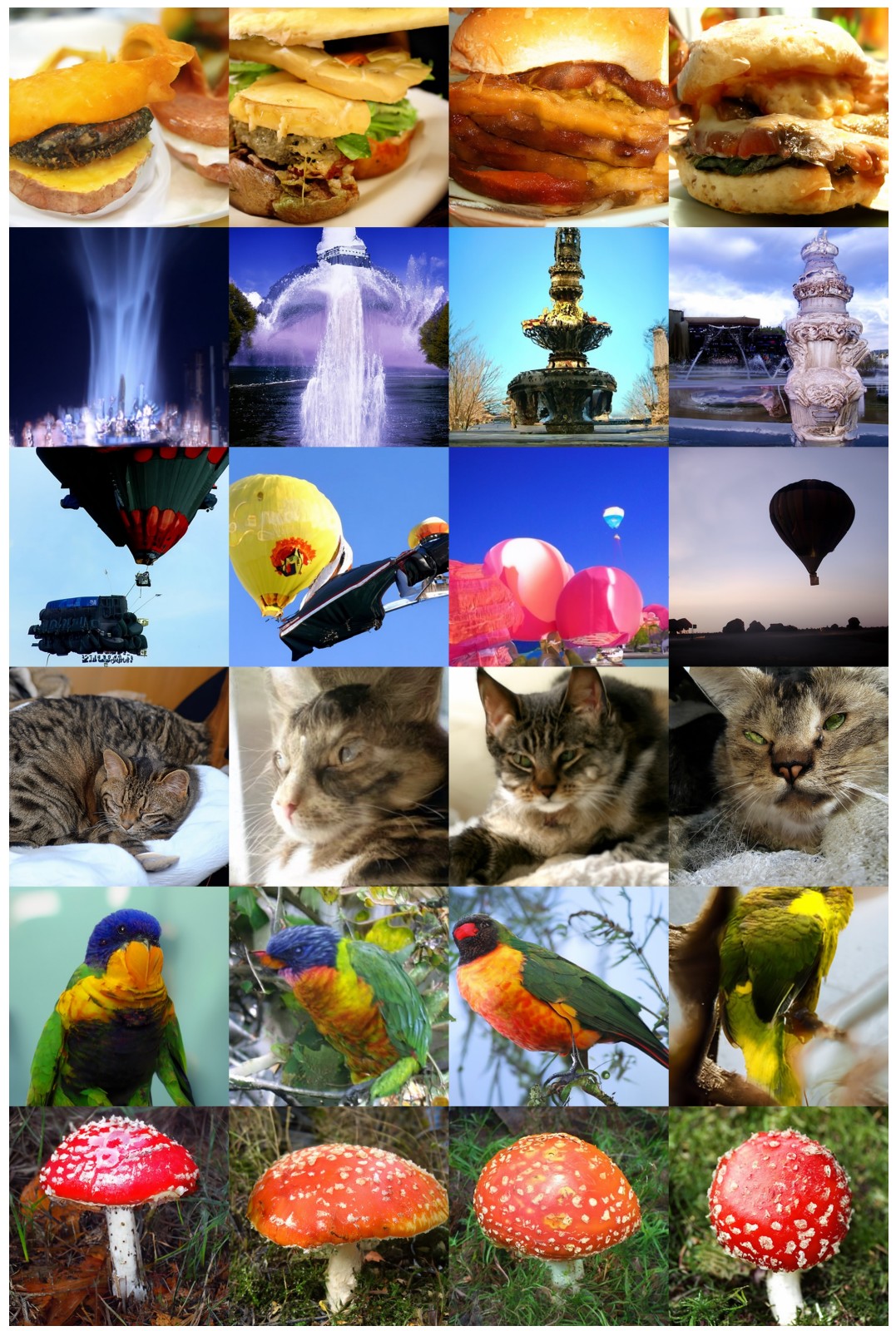

Figure 17: Samples from our guided 512×512 model using 250 steps with classifier scale 4.0 (FID 7.72). Classes are 933: cheeseburger, 562: fountain, 417: balloon, 281: tabby cat, 90: lorikeet, 992: agaric.

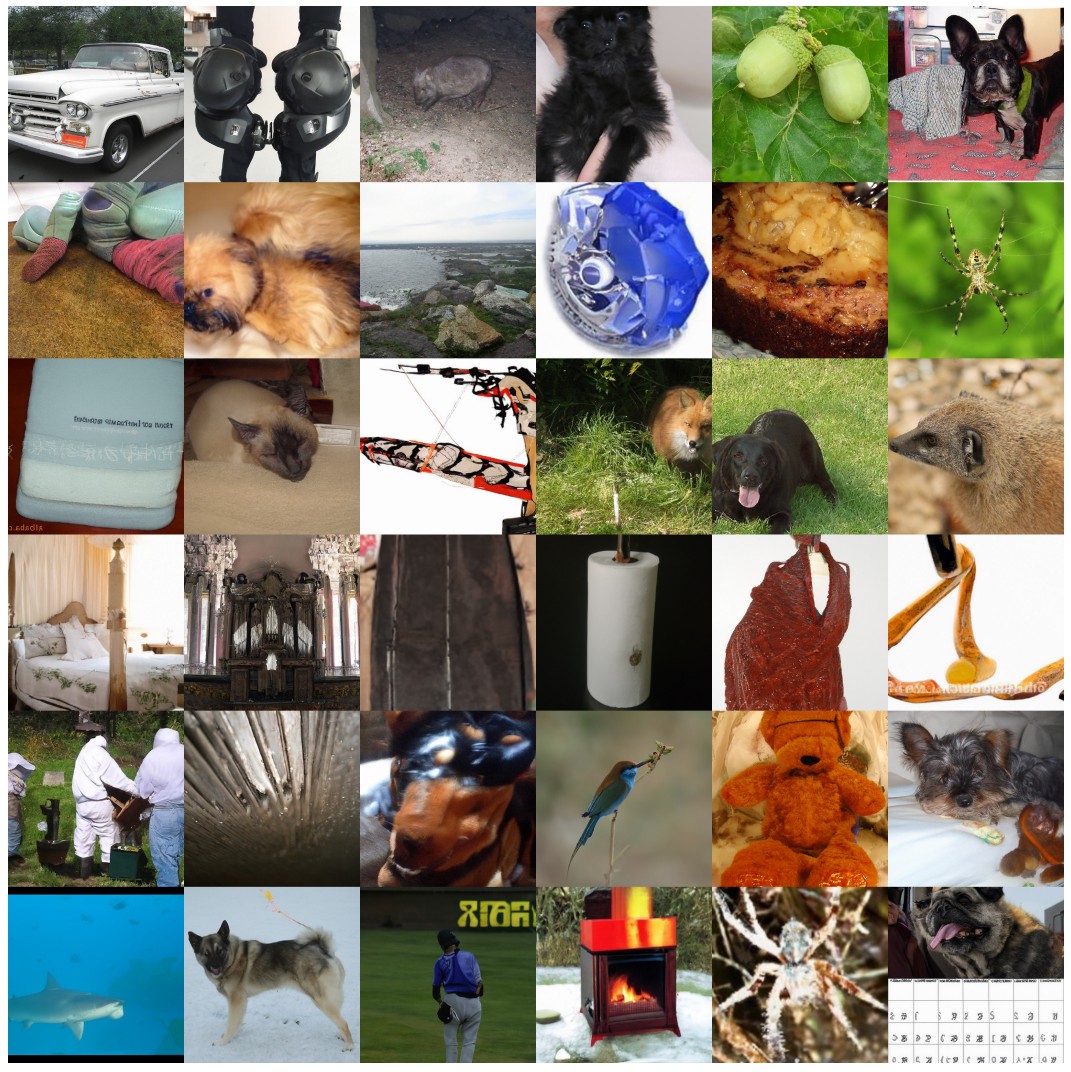

Figure 18: Random samples from our best ImageNet 512×512 model (FID 3.85).

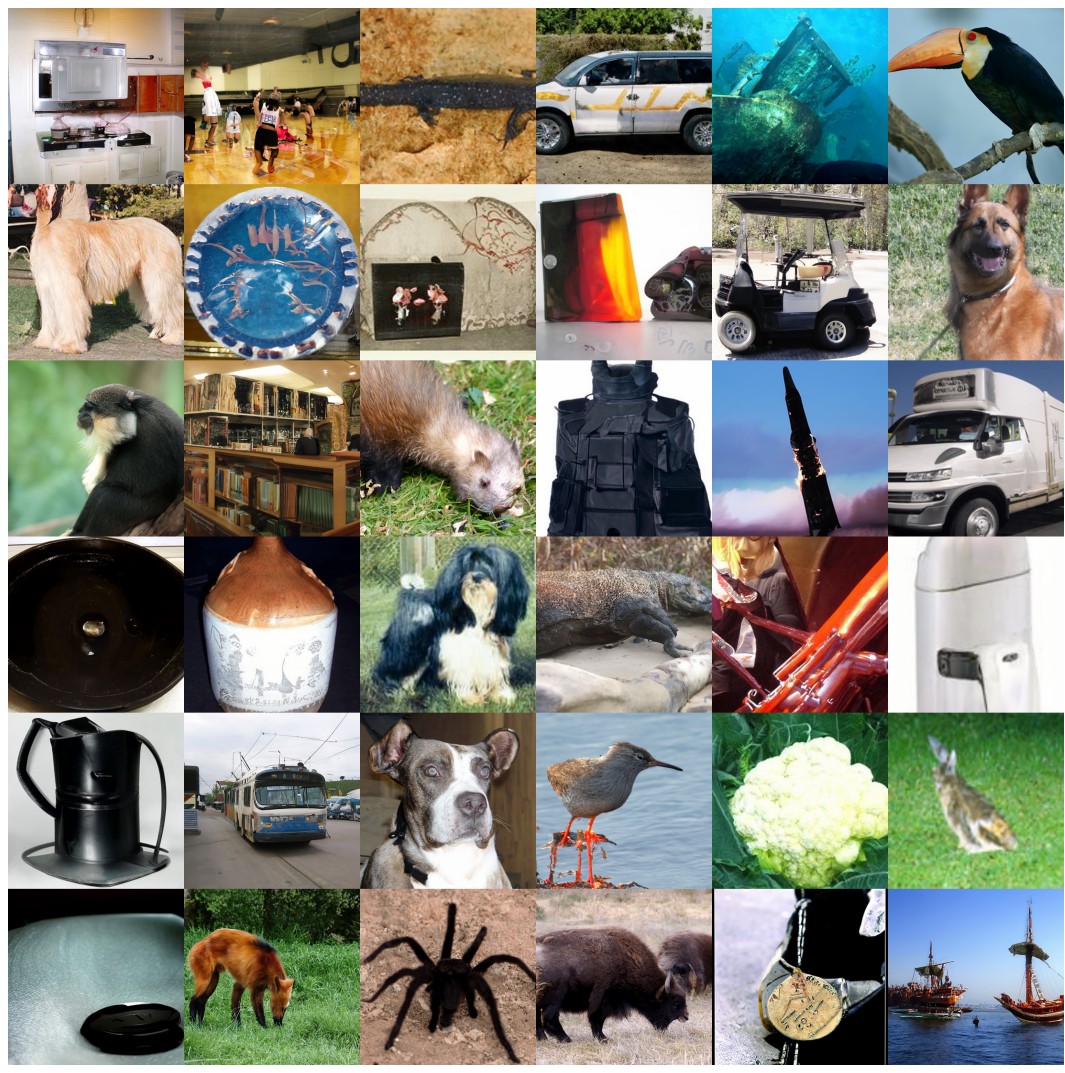

Figure 19: Random samples from our guided 512×512 model using 250 steps with classifier scale 4.0 (FID 7.72).

# N    Samples from ImageNet 256×256

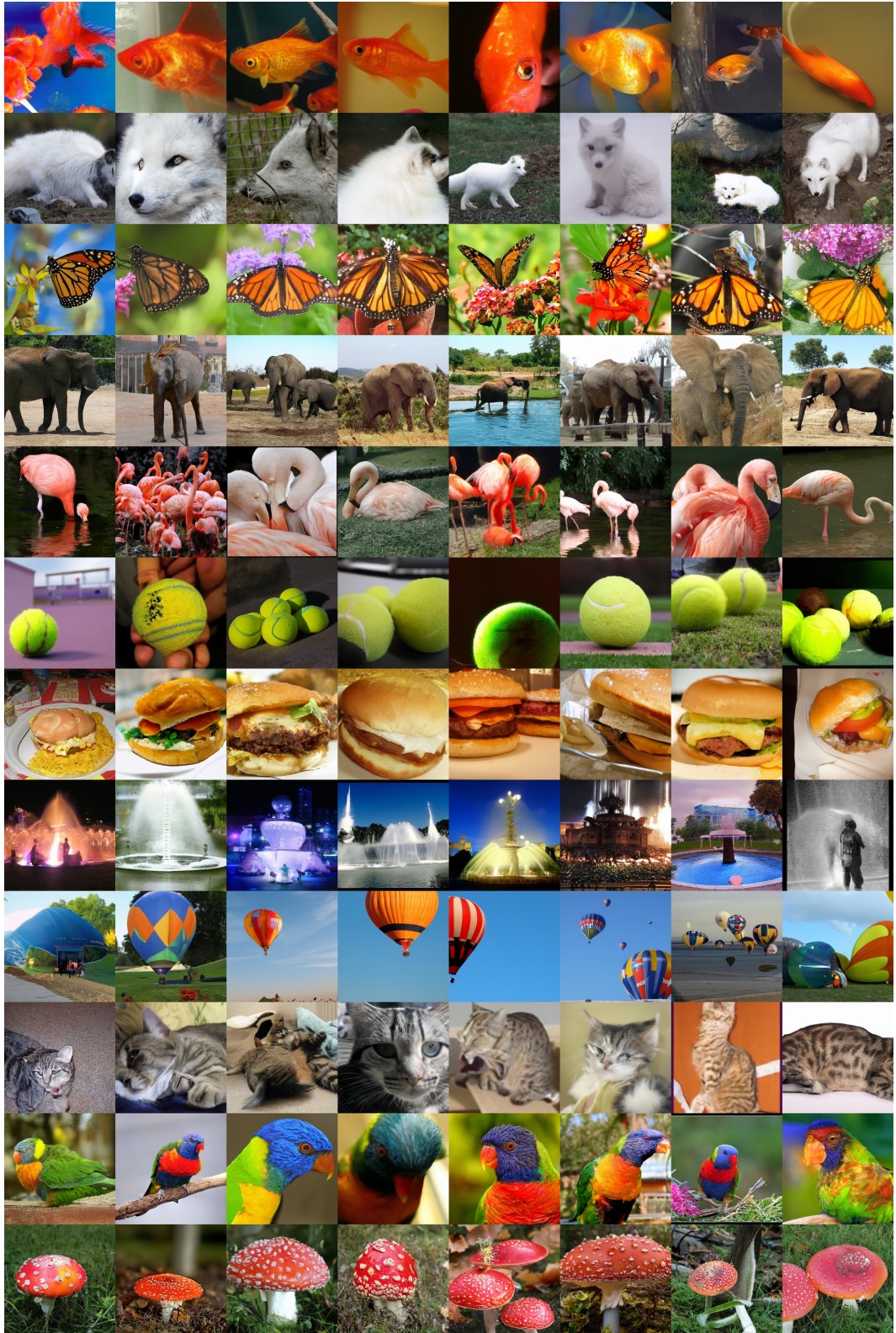

Figure 20: Samples using our best 256×256 model (FID 3.94). Classes are 1: goldfish, 279: arctic fox, 323: monarch butterfly, 386: african elephant, 130: flamingo, 852: tennis ball, 933: cheeseburger, 562: fountain, 417: balloon, 281: tabby cat, 90: lorikeet, 992: agaric

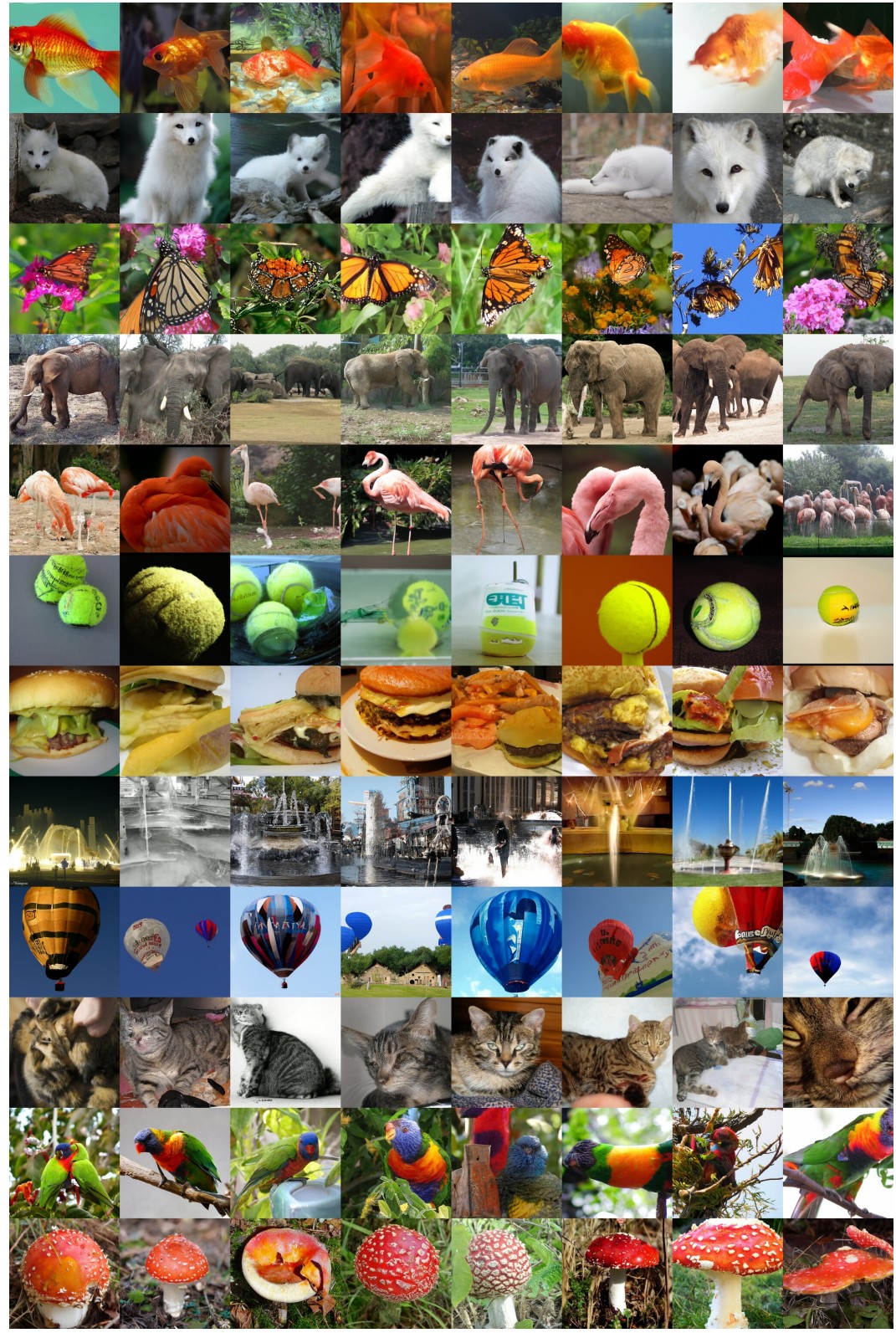

Figure 21: Samples from our guided 256×256 model using 250 steps with classifier scale 1.0 (FID 4.59). Classes are 1: goldfish, 279: arctic fox, 323: monarch butterfly, 386: african elephant, 130: flamingo, 852: tennis ball, 933: cheeseburger, 562: fountain, 417: balloon, 281: tabby cat, 90: lorikeet, 992: agaric

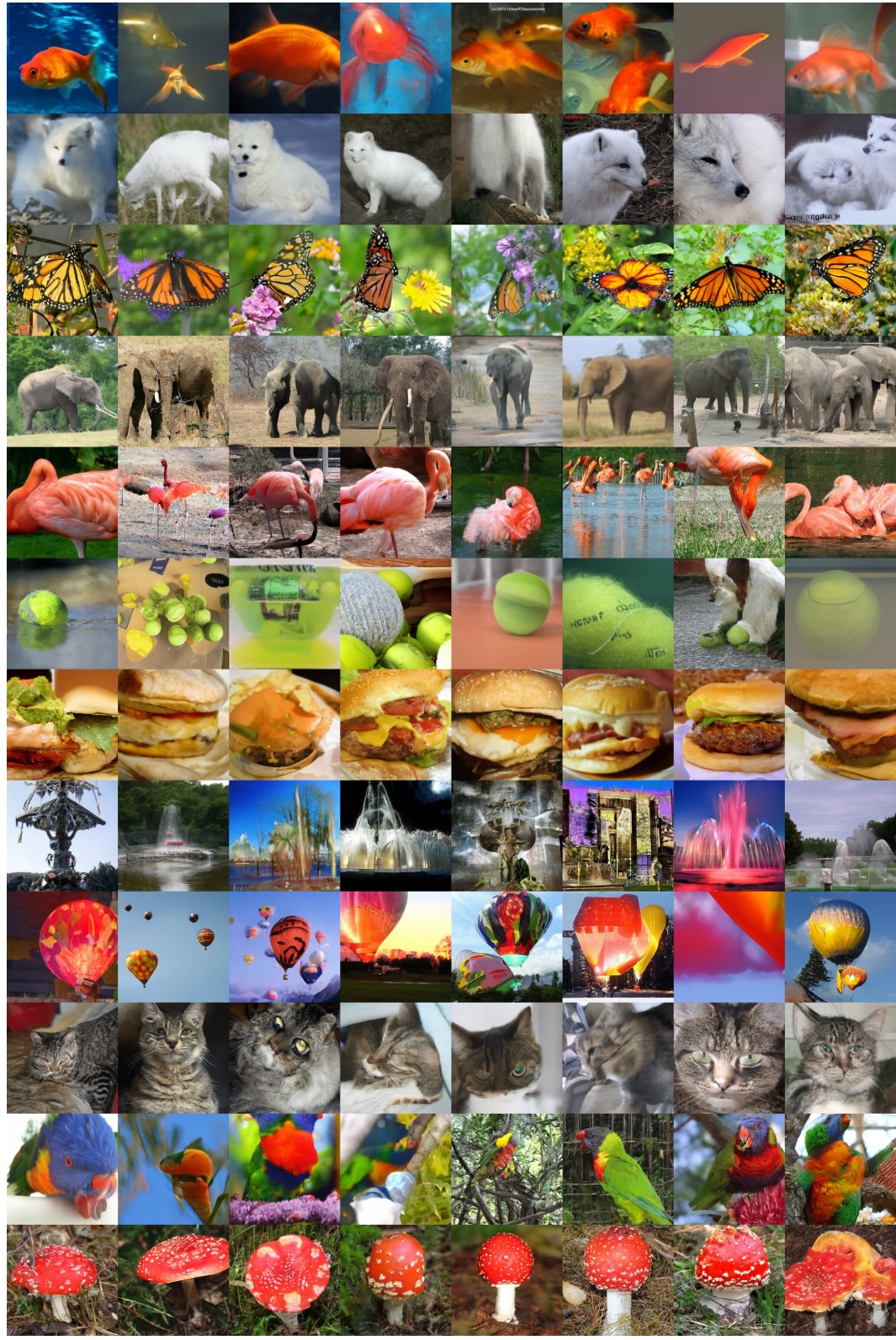

Figure 22: Samples from our guided 256×256 model using 25 DDIM steps with classifier scale 2.5 (FID 5.44). Classes are 1: goldfish, 279: arctic fox, 323: monarch butterfly, 386: african elephant, 130: flamingo, 852: tennis ball, 933: cheeseburger, 562: fountain, 417: balloon, 281: tabby cat, 90: lorikeet, 992: agaric

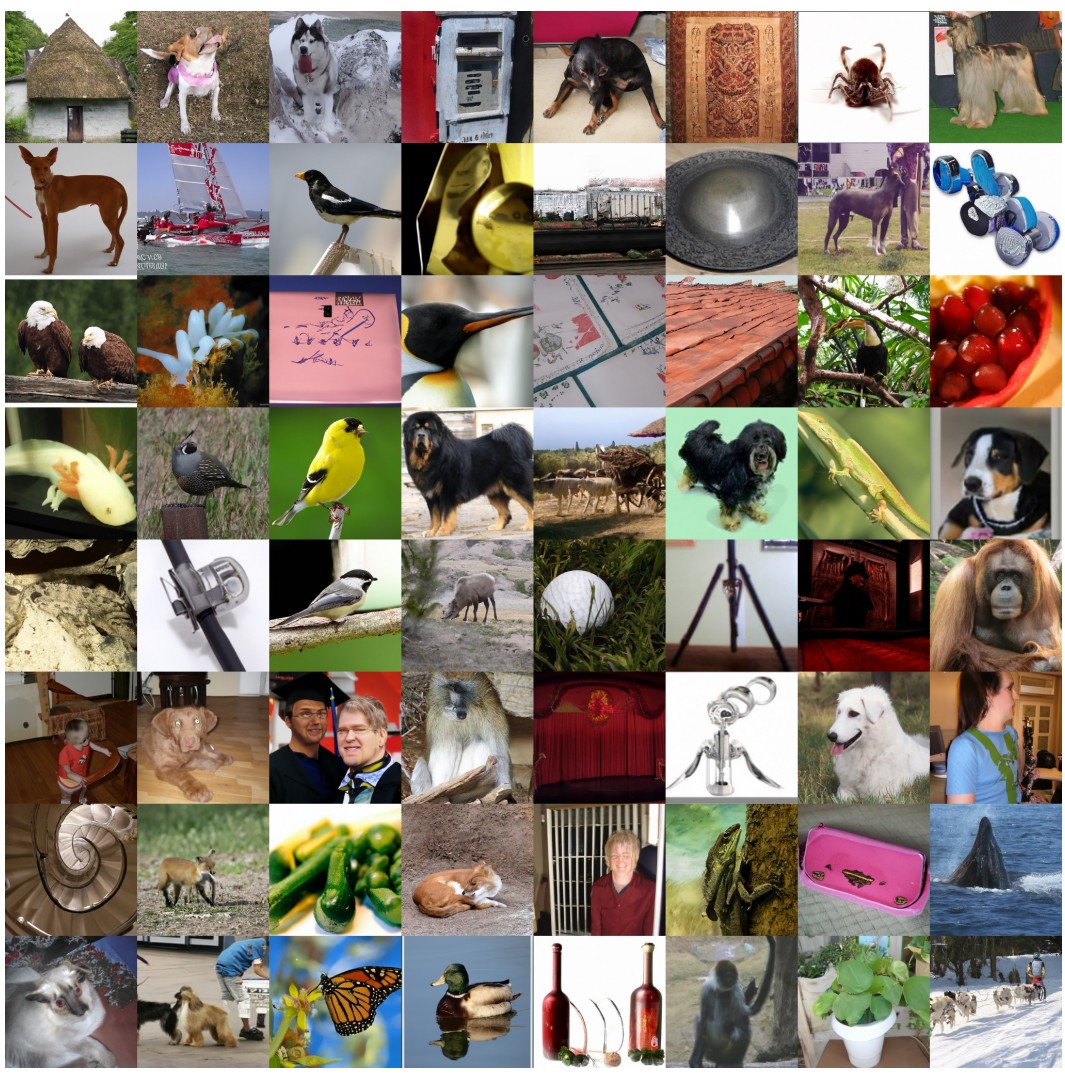

Figure 23: Random samples from our best 256×256 model (FID 3.94).

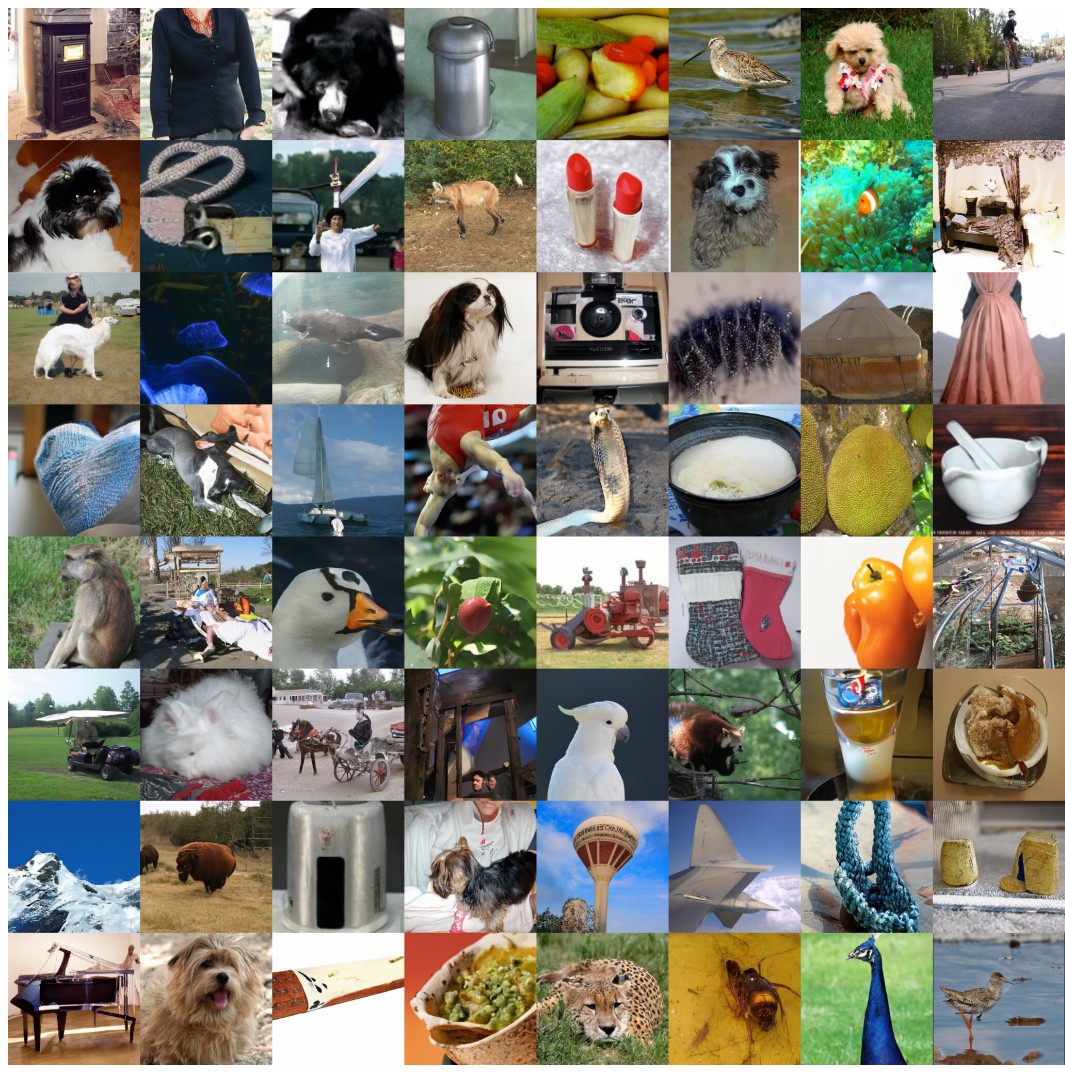

Figure 24: Random samples from our guided 256×256 model using 250 steps with classifier scale 1.0 (FID 4.59).

## O    Samples from LSUN

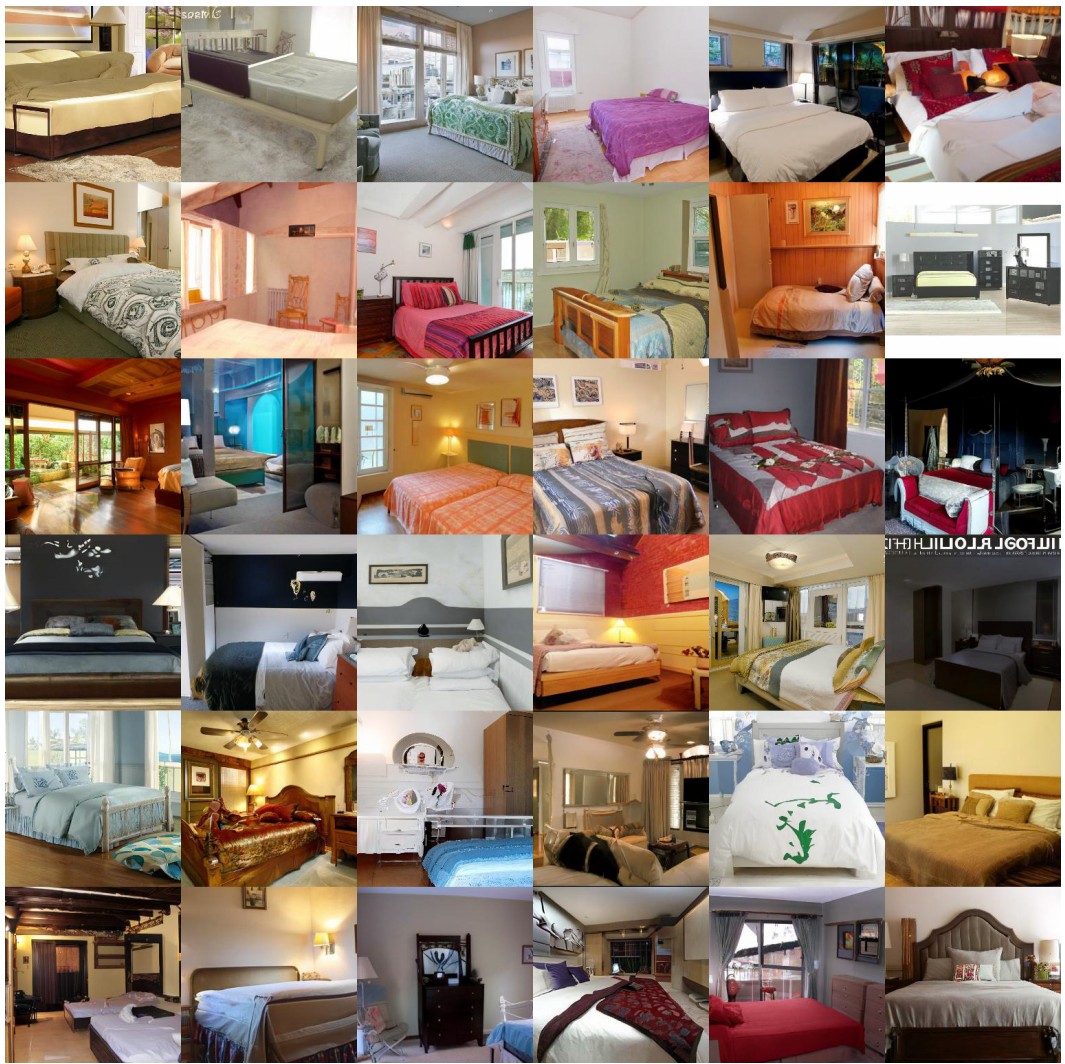

Figure 25: Random samples from our LSUN bedroom model using 1000 sampling steps. (FID 1.90)

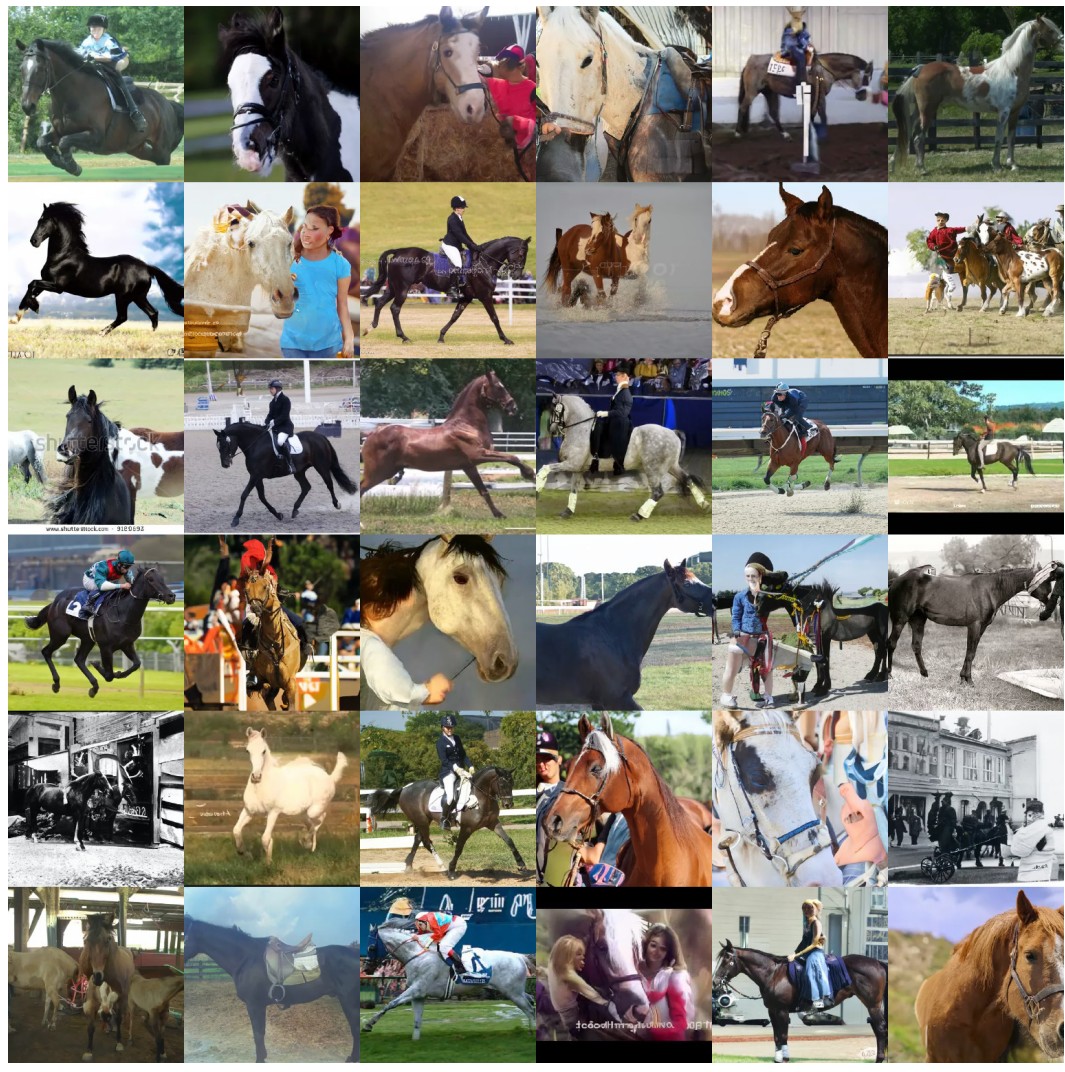

Figure 26: Random samples from our LSUN horse model using 1000 sampling steps. (FID 2.57)

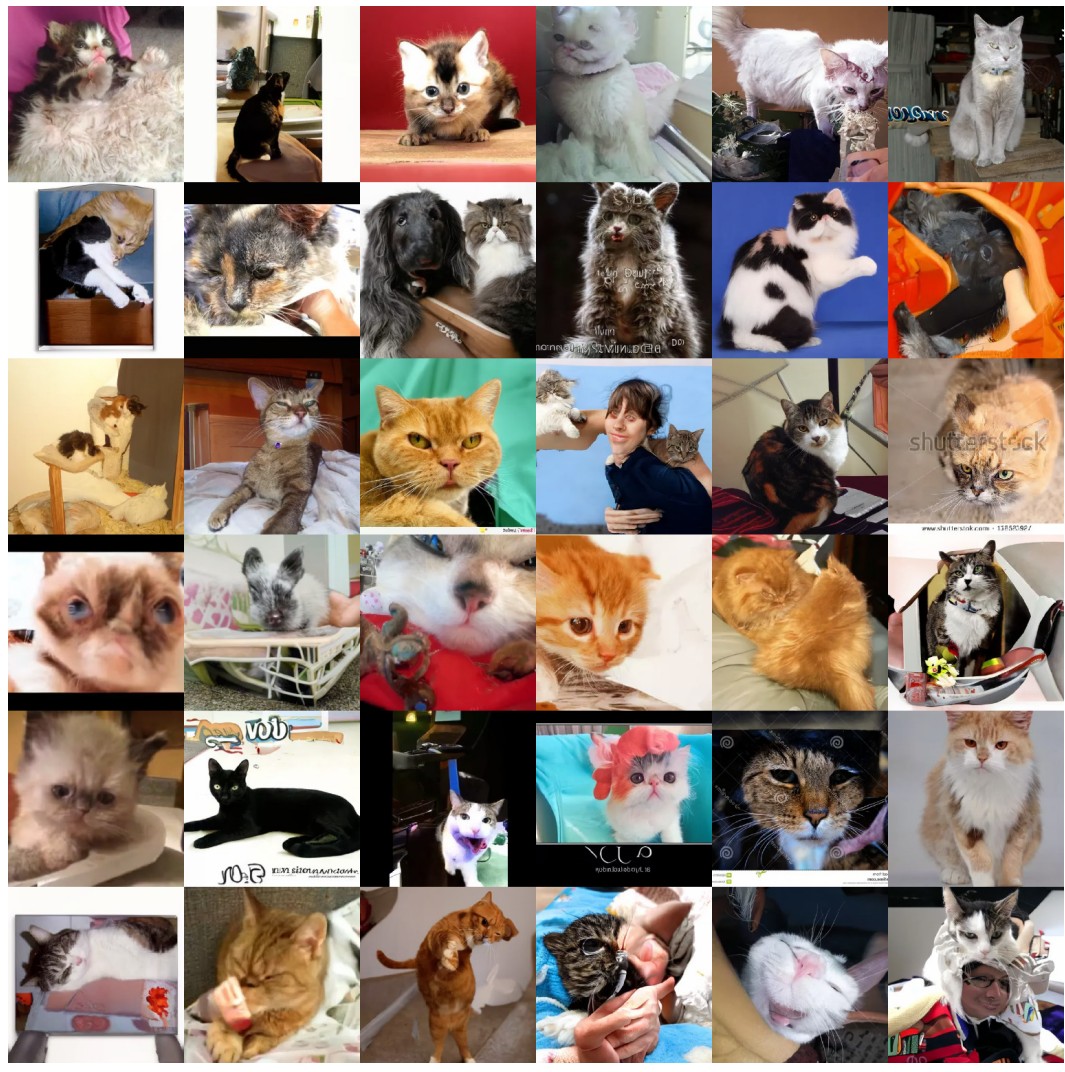

Figure 27: Random samples from our LSUN cat model using 1000 sampling steps. (FID 5.57)