# OpenReview forum: "Diffusion Models Beat GANs on Image Synthesis"
_NeurIPS.cc/2021/Conference — NeurIPS 2021 Spotlight_

### Official Review · Reviewer_H51k · 2021-07-09

**Rating:** 7
**Confidence:** 4

**Summary:**

This work shows that with some architectural changes, classifier guided sampling and comparable compute requirements, diffusion models can achieve state of the art performance in generative image modeling. The paper provides a wealth of empirical data on the performance of diffusion models and how it is influenced by architectural choices and classifier guidance. Ablations on these components lead to FID scores which improve upon GAN based approaches.

**Limitations And Societal Impact:**

Limitations regarding sampling speed and the applicability of classifier guidance to unlabeled datasets are addressed and general, potential societal impacts of generative models are discussed.
Another important limitation that could be discussed is the applicability of diffusion models to image modifications and representations. For example, GANs provide a low-dimensional, well-behaved latent code that can be directly decoded to an image. This provides ways to modify and optimize such codes. While the DDIM sampling process provides a way to obtain a latent code with diffusion models, the interpolations shown in the supplementary look worse than those obtained by BigGAN. Similarly, optimizing these latent codes with respect to some desired image outcome is probably infeasible due to the need to run a large number of forward passes. Another simple and interesting experiment would be to decode the same latent code under different class conditionings to see if the latent code captures class-agnostic information (such as pose etc.) similar to BigGAN.

**Main Review:**

__Significance__
Diffusion models are a timely topic and a promising approach to generative modeling. The paper mostly builds upon [46] and demonstrates that diffusion models can achieve state-of-the-art FID scores for unconditional image synthesis tasks on LSUN and class-conditional image synthesis tasks on ImageNet. This is mostly achieved based on extensive experiments regarding architecture changes and classifier guidance.

__Presentation__
The paper itself is very readable and well structured. It contains the necessary background and is self-contained by providing the model formulation and derivations in the supplementary. Code is included which helps to understand the precise architecture and, potentially, to reproduce results. Unfortunately, I could not find the implementation used to compute FID scores. As pointed out in l.96, implementations of FID evaluations can differ in many subtle ways making comparisons difficult without access to the precise implementation used.

__References__
The paper covers many relevant works on improving diffusion models and score based models. However, given the importance of evaluating the performance on class conditional ImageNet modeling, the related works in this area should be covered more completely. In particular, Esser et al. "Taming Transformers for High-Resolution Image Synthesis." also rely on a more intelligent compression scheme like DCTransformer [45] and report better results, Devries et al. "Instance Selection for GANs." also greatly reduce the computational resources while maintaining performance of BigGAN [8], and Fauw et al. "Hierarchical Autoregressive Image Models with Auxiliary Decoders." use a hierarchical approach related to the upsampling approach used in [46] and [56].

__Strengths__
- The paper demonstrates very impressive generative performance of diffusion models on a variety of datasets and resolutions.
- It provides evidence that diffusion models can be scaled to complex and high-resolution image synthesis tasks.
- It provides a wealth of useful empirical data on the influence of architectural changes and classifier guidance (and its scale) on the performance of diffusion models.

__Weaknesses__
- The paper contains no methodological novelty. The architecture and training mostly follow [46] (and [8,63] regarding the residual block); classifier guidance follows [59].
- It provides limited additional insights. [46] already demonstrated improved performance of diffusion models over BigGAN on ImageNet at resolution 64x64, and even how performance improves in a predictable manner with increasing model sizes. The only additional insights over [46] are thus that (i) performance can be further increased with hyperparameter search regarding the architecture and the additional use of classifier guidance, and (ii) the same holds for higher resolutions.

__Comments and Suggestions__
- Will the pre-trained models be released? Given that the compute resources that went into training the models is one of the main contributions of this work, and the fact that there are still many questions to be explored for diffusion models, availability of these models would increase the significance of this work to other researchers and lead to further progress.
- l.14: What exactly is the ability to generate "infinite" high-quality synthetic images?
- l.62: sqaured -> squared
- l.98: It would be helpful to provide the code used for FID evaluation so future works do not run into the same issue of subtle differences which requires re-evaluation of previous works and becomes impossible if neither models or samples are available.
- l.103: colvolutions -> convolutions
- l.636: how is the class for the interpolated latents choosen when decoding them?

__Rating__
Overall, I think that the results are impressive and that the work will serve as an important reference on what is achievable with diffusion models. On the other hand, the work provides rather limited novelty and insights on diffusion models beyond previous works such as [46] which leads to my rating of a weak accept.


**Time Spent Reviewing:**

8

---

> ### Author Response · Authors · 2021-08-09
> **Small edits, emphasize novelty, address latent space**
>
> Thank you for your thoughtful comments which have helped us improve our paper!
>
> - We have removed the term “infinite” from this sentence, as it has some incorrect / problematic implications.
> - We have added references to Instance Selection GAN and “Hierarchical Autoregressive Image Models with Auxiliary Decoders”. We also evaluated VQGAN using our evaluation pipeline and have added this to our results table.
> - Regarding novelty: while [59] proposes classifier guidance as a means of adding conditioning to an unconditional model, they do not suggest that it could improve sample quality when used on top of an already class-conditional model, or that the classifier gradients could be scaled to trade-off diversity for fidelity. We consider this a novel contribution which was not obvious before our paper and provides a huge boost in quality that had been previously untapped.
> - Pre-trained models will indeed be released!
> - We mention in Appendix J that we use the original code from TTUR [26] directly to compute FID without resizing images beforehand. With our released models, we will also release our sample and reference batches, and a script to compute our evaluation metrics on them. We hope that future papers will switch to Clean FID (https://www.cs.cmu.edu/~clean-fid/) or similar, and releasing our models+samples should aid future researchers in this transition.
> - We have fixed the typos you discovered. Thanks for your attention to detail!
> - In Appendix H, we interpolate between random samples from the dataset, using their corresponding class labels. We have added the phrase “randomly selected” before “ground truth dataset examples” to clarify this.
> - Regarding the usefulness of latent space: it indeed appears that the DDIM latent space is not great for interpolation, and we have not explored editing it in this paper. In retrospect, it is unclear if a good latent space actually matters for image editing applications, since recent work achieved high-quality diffusion-based image editing (https://arxiv.org/abs/2108.01073). Nevertheless, we have added a paragraph to the limitations section of our paper regarding the lack of learned latent representations.

---

> > ### Comment · Reviewer_H51k · 2021-08-22
> > **Raising Score to 7**
> >
> > Thanks for addressing my concerns. I agree that the effects of classifier guidance on sample quality and diversity have not been explored before. In combination with the release of pre-trained models, I think this work will benefit future research on generative models and I therefore raise my score from a 6 to a 7.

---

### Official Review · Reviewer_rkcu · 2021-07-13

**Rating:** 7
**Confidence:** 4

**Summary:**

The authors demonstrate that through architectural improvements, diffusion models can achieve state-of-the-art image sample quality, beating previous GANs that have dominated the image generation task for years. They also demonstrate that the sample quality can be further improved post-training using classifier guidance [1,2].

[1] Deep unsupervised learning using nonequilibrium thermodynamics (Sohl-Dickstein et al. 2015)
[2] Score-based generative modeling through stochastic differential equations (Song et al. 2020)



**Limitations And Societal Impact:**

Limitations and Societal Impacts are adequately discussed in Sections 7 and 8, respectively.

**Main Review:**

The main focus of the paper is demonstrating the capabilities of diffusion models for image synthesis. The paper does a great job at conveying this message and is very clearly written. It is interesting to see that after less than a year of research interest around diffusion models, they can beat GANs which have dominated the image synthesis task for years.

The experiments are strong, containing an ablation study on the neural architecture, a study on the effect of classifier guidance on sample quality, as well as a larger-scale study using metrics beyond just FID, including sFID [2] and Precision and Recall [3] where state-of-the-art performance is demonstrated on several image datasets.

In terms of technical innovation, the paper has few contributions. The differences compared to [1] are mostly about scaling up parts of the model and using previously proposed changes, which is studied in the ablation study. This is however fine as it is not the main focus of the paper.

One point of potential improvement could be to include a short discussion / a column in Table 1 on the demands on computation/memory/parameter count as various model components are ablated.

[1] Improved Denoising Diffusion Probabilistic Models (Nichol & Dhariwal, 2020)
[2] Generating images with sparse representations (Nash et al., 2021)
[3] Improved Precision and Recall Metric for Assessing Generative Models (Kynkäänniemi et al., 2019)

**Time Spent Reviewing:**

3

---

> ### Author Response · Authors · 2021-08-09
> **Address novelty, reference wall-clock time figures**
>
> Thanks for providing feedback and taking the time to review our work!
>
> - Regarding novelty: the main novel contribution beyond our systematic ablations is using classifier guidance to trade-off diversity for fidelity. We are not aware of any previous paper that has applied classifier guidance to models which are already class-conditional, nor have any papers explored rescaling the classifier gradients. These ideas are essential to our results, and it was not obvious before writing this paper that using classifier guidance in this way had any reason to boost quality or allow us to trade-off diversity for fidelity.
> - In Appendix A, we look at wallclock time for these architecture changes. For the most part, we aimed to ablate the architecture while holding compute/parameters/memory constant, but the compute time doesn’t end up being exactly the same (e.g. GPUs are slower when evaluating deeper, narrower networks even with the same theoretical FLOP count).

---

> > ### Comment · Reviewer_rkcu · 2021-08-18
> > **Thanks**
> >
> > Thanks a lot for your reply.
> > I believe the strong results and the insights on how to achieve them warrant acceptance.

---

### Official Review · Reviewer_NiCt · 2021-07-16

**Rating:** 7
**Confidence:** 4

**Summary:**

The authors explore advances in large scale diffusion models for image generation, and with guidance from an image classifier gradient, demonstrate that diffusion models can outperform state of the art GAN models on image synthesis at different resolutions with respect to standard metrics such as Inception Score, FID. The Classifier guidance also offers a trade-off between sample quality and diversity for diffusion models.

**Limitations And Societal Impact:**

The authors adequately address the limitations and potential negative societal impact of their work.



**Main Review:**

- The authors take existing diffusion models, and classifier guidance techniques, and along with few architecture modifications motivated from previous works and show that such a combination on large scale can beat state of the art GAN based models.
- The paper is clearly written and easy to follow. The experiments are well conducted, with detailed ablation studies.
- The paper can be of significance to the generative modeling community in establishing the capability of large scale likelihood based diffusion models on the challenging task of image generation.
- Few questions:
  - I will be interested to compare the degree of memorization of the model as a function of the scale factor of the classifier guidance.
    The authors do show some handful samples for scale factor 1.0 and 2.5, but it would be nice to use some form of automated
    metric (e.g. minimum or average min-K Inception distance between each image and dataset). It might be the case, that for some
    classes, memorization is more likely than other classes. Also what happens at a scale factor of 10, does the model start
    memorizing at these scale factors?
  - Did the authors explore the benefit of classifier guidance as a function of classifier's performance on the validation set? In general
     how strong is the classifier that the authors trained using the UNet backbone?

**Time Spent Reviewing:**

3

---

> ### Author Response · Authors · 2021-08-09
> **Discuss memorization, high classifier scales, classifier accuracy**
>
> Thank you for taking the time to study our work and provide thoughtful feedback!
>
> - Quantifying memorization is, in many ways, an open problem in generative modeling. The standard practice for image generation papers is to visually inspect a handful of nearest neighbors, but indeed this doesn’t allow you to quickly check all of the classes. Precision and recall are both closely related to the min-K metric you proposed, although they are typically computed over 10k training images instead of the whole dataset due to compute constraints. Doing a large-scale nearest neighbor search might shed some insight, especially if it was adopted as a standard metric, although it is actually a non-trivial engineering problem requiring a possibly large amount of compute.
> - Generally, increasing classifier scale to high values produces somewhat unrealistic, strangely washed-out images (even though Inception Score keeps improving). This seems to suggest that the classifier isn’t necessarily helping the model memorize individual training images, since in the limit the images don’t look realistic but do contain many class-specific visual features.
> - We have not deeply explored how classifier performance affects sample quality. One datapoint we have here is that training classifiers for much less time doesn’t significantly hurt sample quality, as we find in Appendix B.3 Table 10. To give a sense of accuracy, the 150k and 500k iteration classifiers get 60.7% and 64.1% top-1 accuracy, respectively, when measured at timestep 0 (no noise). It is hard to get a sense for how good these classifiers are in an absolute sense, since they are trained on a much harder problem (i.e. noisy inputs) than traditional ImageNet classifiers.

---

> > ### Comment · Reviewer_NiCt · 2021-08-29
> > **Thank you for the response.**
> >
> > I thank the reviewers for answering my questions. As I said in my main review, the paper can be of significant importance to the generative modeling community, and hence I keep my rating unchanged, and vote for acceptance.

---

### Official Review · Reviewer_Fjfm · 2021-07-24

**Rating:** 9
**Confidence:** 5

**Summary:**

This work demonstrates that diffusion models can outperform the state-of-the-art GAN models on image synthesis.  On unconditional image synthesis, the authors find a better architecture through a series of ablations. On conditional tasks (conditioned on class labels), the authors trade off diversity for fidelity using gradients from a classifier the same architecture.

**Limitations And Societal Impact:**

Yes.

**Main Review:**

Originality:
- Most of the ideas in this paper are not brand new and were proposed by previous work (e.g., classifier guidance). However, the authors put them together and execute them really well.

Quality:
- The submission is technically sound.

Clarity:
- This paper is well-written.

Significance:
- This paper contains a good number of insights on building the state-of-the-art diffusion models for image synthesis.
- The architecture improvements are significant.

Comments:

The classifier uses the same architecture as the diffusion model. Is there any special reason? Did the authors try different architecture for classifier guidance, e.g., ResNet with time-step embedding?

Overall, this paper provides successful recipes and non-trivial insights for building state-of-the-art diffusion models for image synthesis. I think it is a good paper. I enjoyed reading it.

**Time Spent Reviewing:**

8

---

> ### Author Response · Authors · 2021-08-09
> **Discuss architecture choices and initial exploration**
>
> Thank you so much for taking the time to read our paper and provide feedback!
>
> Regarding classifier architecture: we did briefly experiment with WideResNet-50 classifiers (with timestep embedding) and various different kinds of pooling layers on the UNet. We opted not to use the ResNet because it appeared much worse at handling noise (it’s average training accuracy was much lower than the UNet encoders as a result). We chose attention pooling in the UNet because of a slight boost in validation accuracy.

---

### Public Comment · ~Ernie_Chu1 · 2023-07-22
**An error in Algorithm 2**

Hi, thank you for the insightful work. I have a minor question toward Algorithm 2. In classifier guided DDIM sampling, it seems like the gradient scale $s$ was never used. Did I miss something?

---

### Decision · Program_Chairs · 2021-09-28

**Decision:**

Accept (Spotlight)

**Comment:**

All reviewers agree that this is a well-written paper with strong experiments that provide non-trivial insights and results in state-of-the-art diffusion models for image synthesis. The evaluation goes beyond prior work in terms of evaluating sFID, Precision-Recall, and perorming several ablations on neural architecture and the impact of classifier guidance on sample quality. While there were some concerns around the novelty of proposed methods, the careful experiments and strong empirical results in this paper helps to advance the capabilities of generative image models in an impressive fashion.

**Consistency Experiment:**

NeurIPS has a long history of experimentation. In 2014, NeurIPS ran an experiment in which 10% of submissions were reviewed by two independent committees to quantify the randomness in the review process. This year, we repeated a variant of this experiment to see how the quality of the review process has changed over time.  This paper was part of the experiment and was therefore assigned to two committees (consisting of reviewers, an Area Chair, and a Senior Area Chair) that reached independent decisions.  If both committees made the same recommendation, this recommendation was followed. If a single committee recommended acceptance, the paper was accepted (with the exception of a few cases in which the other committee identified what we considered a fatal flaw, e.g., an error in a key result).

Both committees reached the same decision: **Accept (Spotlight)**

The other committee assigned to the paper recommended **Accept (Spotlight)**.  You can find the other set of reviews, along with any follow up discussion with the authors here:
https://openreview.net/forum?id=OU98jZWS3x_